# MDM2 functions as a timer reporting the length of mitosis

**Luke J. Fulcher, Tomoaki Sobajima ⬥ , Caleb Batley, Ian Gibbs-Seymour ⬥ & Francis A. Barr ⬥ ✉**

Delays in mitosis trigger p53-dependent arrest in G1 of the next cell cycle, thus preventing repeated cycles of chromosome instability and aneuploidy. Here we show that MDM2, the p53 ubiquitin ligase, is a key component of the timer mechanism triggering G1 arrest in response to prolonged mitosis. This timer function arises due to the attenuation of protein synthesis in mitosis. Because MDM2 has a short half-life and ongoing protein synthesis is therefore necessary to maintain its steady-state concentration, the amount of MDM2 gradually falls during mitosis but normally remains above a critical threshold for p53 regulation at the onset of G1. When mitosis is extended by prolonged spindle assembly checkpoint activation, the amount of MDM2 drops below this threshold, stabilizing p53. Subsequent p53-dependent p21 accumulation then channels G1 cells into a sustained cell-cycle arrest, whereas abrogation of the response in p53-deficient cells allows them to bypass this crucial defence mechanism.

Chromosome instability, aneuploidy, the removal of centrosomes or antimitotic drugs targeting the microtubule cytoskeleton delay progression through mitosis, triggering a p53- and p21-dependent cell-cycle arrest in G1 thought to prevent proliferation of damaged cells[1–6]. This response is lost in cancer cells in which p53 has become inactivated by mutation or other mechanisms, including expression of viral oncoproteins[7–10]. Crucially, G1 cell-cycle arrest following prolonged mitosis occurs even in the absence of detectable DNA damage, suggesting it has a different cause, proposed to be a direct consequence of the increased time spent in mitosis[1–10]. Because most studies have focussed on cells lacking centrosomes and drug-induced mitotic delays, the role of this pathway and the molecular components required for normal cell function remains unclear. Therefore, we asked if variation in the length of mitosis, inherent in the stochastic search-capture process underpinning chromosome alignment even in the absence of any perturbation, influences the behaviour of untransformed diploid cells with wild-type p53 (p53^WT) in the ensuing G1.

## Results

### Delays in mitosis triggers p53-dependent arrest in G1
We used fluorescence imaging and single cell tracking of telomerase-immortalized retinal pigmented epithelium cells (hTERT-RPE1) stably expressing a FUCCI reporter to track cell-cycle progression[11].

For individual cells under normal growth conditions, we measured the length of mitosis (M) from nuclear envelope breakdown, and subsequent G1 duration and recorded whether G1 cells entered S-phase (Fig. 1a–c). The majority of p53^WT cells tracked with this method completed mitosis in under 60 min, with a mean time of 50.3 ± 9.7 min (Fig. 1b). All cells spending 40–49 min in mitosis passed through G1 and entered S phase in 10.6 ± 2.7 h (Fig. 1a,c). For cells spending 50–59 min in mitosis, G1 length was increased to 13.4 ± 7.3 h for cells entering S phase, and 19% ± 7% of cells arrested in G1 (Fig. 1a,c). By contrast, 75% ± 3% of p53^WT cells that spent more than 60 min in mitosis arrested in G1 for at least 27.9 ± 10.0 h without any notable increase in cell death (Fig. 1a,c). For hTERT-RPE1 p53-knockout (p53^KO) cells, the mean time in mitosis was 51.0 ± 11.7 min, not significantly different to p53^WT cells (Fig. 1b), showing p53 does not play a direct role in mitotic timing. However, unlike p53^WT cells, neither increased G1 length nor cell-cycle arrest were observed in p53^KO cells spending longer than 60 min in mitosis (Fig. 1a,c). Crucially, both p53^WT and p53^KO cells have normal centriole numbers and are able to form primary cilia in G1 when deprived of growth factors (Extended Data Fig. 1a,b), suggesting the differences we observed in cell cycle progression following delays in mitosis are not due to alterations in centrosome function. Thus, there is a threshold time in mitosis beyond which cells undergo p53-dependent arrest in G1 of the subsequent cell cycle.

Department of Biochemistry, University of Oxford, Oxford, UK. ✉e-mail: francis.barr@bioch.ox.ac.uk

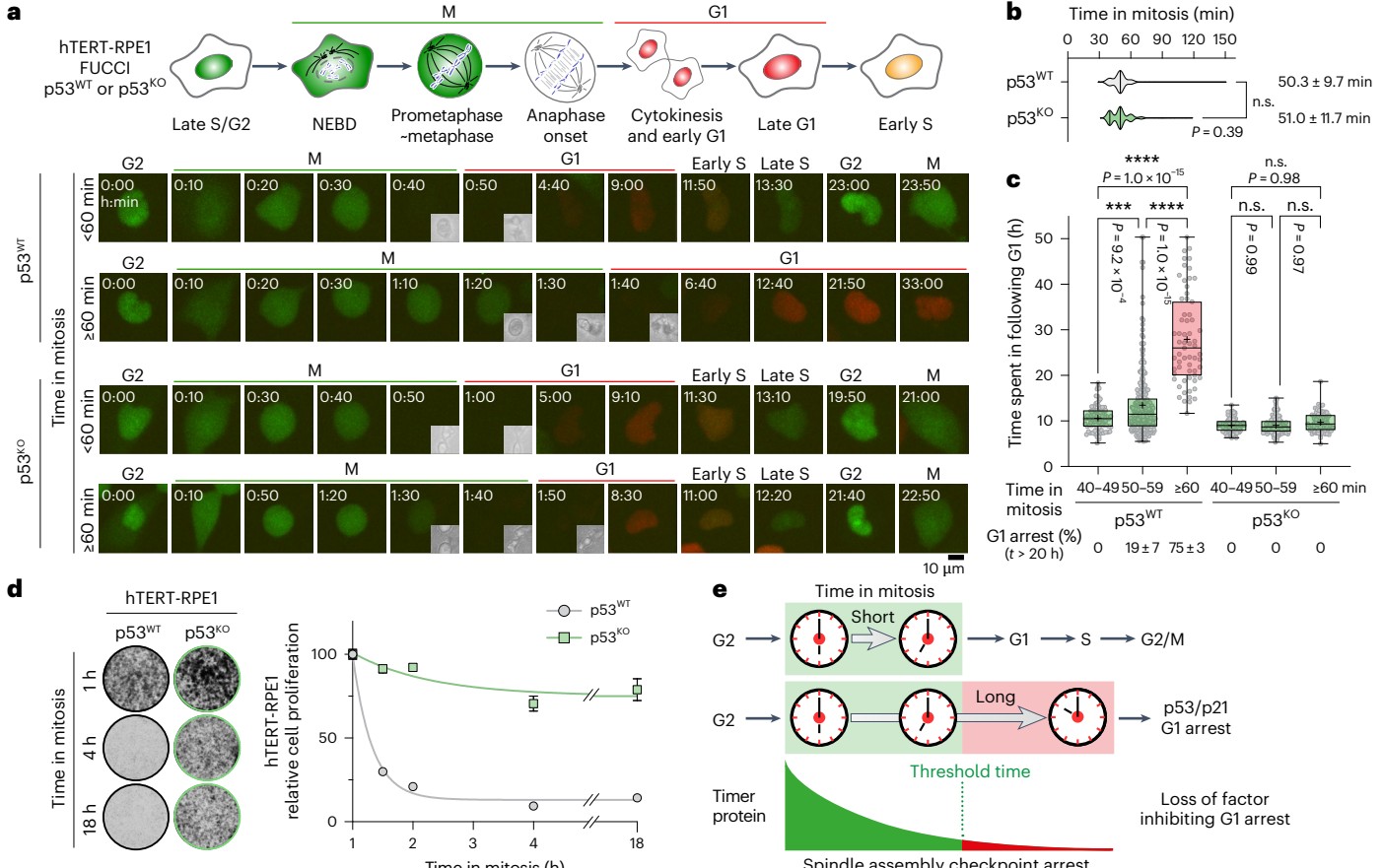

**Fig. 1 | Stochastic variation in mitosis beyond a defined time threshold triggers p53-dependent cell-cycle arrest in the ensuing G1. a,** A schematic of the hTERT-RPE1 FUCCI cell line with representative images of asynchronous cultures of hTERT-RPE1 p53$^{WT}$ and p53$^{KO}$ FUCCI cells entering and exiting mitosis and progressing into the following cell cycle, grouped according to time in mitosis. Cells in G2 were identified, then tracked and imaged every 10 min up to 60 h post mitosis. The insets show anaphase onset and cytokinesis. NEBD, nuclear envelope breakdown; M, mitosis. **b,** The length of mitosis is shown in violin plots (mean ± s.d.; $n_{p53WT}$ = 405 and $n_{p53KO}$ = 286 cells from three independent experiments per cell line; p53$^{WT}$ versus p53$^{KO}$, P = 0.39). **c,** The duration of G1 for p53$^{WT}$ cells spending 40–49 (n = 85 cells), 50–59 (n = 257 cells) or ≥60 min (n = 63 cells) in mitosis or for p53$^{KO}$ cells spending 40–49 (n = 87 cells), 50–59 (n = 131 cells) or ≥60 min (n = 68 cells) is shown in a box and whiskers plot showing the mean (+), median, the 25th and 75th percentiles and whiskers extending to the minimum and maximum values. The percentage of cells arrested in G1 is also shown underneath the plot (mean ± s.d.; n = 3 independent experiments per cell

line) (p53$^{WT}$ cells: P = 9.2 × 10$^{-4}$ for 40–49 versus 50–59 min, P = 1.0 × 10$^{-15}$ for 50–59 versus ≥60 min, P = 1.0 × 10$^{-15}$ for 40–49 versus ≥60 min; p53$^{KO}$ cells: P = 0.99 for 40–49 versus 50–59 min, P = 0.97 for 50–59 versus ≥60 min, P = 0.98 for 40–49 versus ≥60 min) for **a–c. d,** hTERT-RPE1 p53$^{WT}$ or p53$^{KO}$ cells were arrested in mitosis for 1–18 h with 25 ng ml$^{-1}$ nocodazole. The mitotic cells were collected and washed out from nocodazole, and 5,000 cells were plated per well. The cell proliferation after 5 days (mean ± s.e.m.; n = 12 biological samples per cell line from three independent experiments) is plotted in a line graph as a function of time in mitosis for p53$^{WT}$ or p53$^{KO}$ cells. The representative crystal violet-stained wells for 1, 4 and 18 h are shown from three independent experiments with similar results. **e,** A mitotic timer pathway triggers p53-dependent arrest in G1 following delayed mitosis. Loss of a factor inhibiting G1 arrest could act as a sensor for the length of mitosis in this timer pathway. The data were analysed using an unpaired two-tailed t-test with Welch's correction for **b** and a one-way ANOVA test with Tukey's multiple comparisons for **c** (***P < 0.001, ****P < 0.0001).

To more directly test for a link between time in mitosis and G1 cell-cycle arrest, we used transient exposure to a low dose of the microtubule poison nocodazole to perturb chromosome alignment and activate the spindle assembly checkpoint for up to 18 h, similar to the conditions used in earlier studies[1–10]. Checkpoint-arrested mitotic cells were collected and then replated in fresh growth medium lacking nocodazole and tested for proliferation. It is important to note that the low dose of nocodazole arrests cells in mitosis with complete spindles and few unaligned chromosomes (Extended Data Fig. 1c). When the drug is washed out, these chromosomes efficiently align and cells rapidly progress into anaphase, without chromosome segregation errors, and then into G1 to give normal shaped nuclei (Extended Data Fig. 1d). Cell proliferation was still observed after 1 h in mitosis (Fig. 1d), consistent with the FUCCI imaging assays showing this time remains below the threshold needed to trigger G1 arrest. Prolonging mitosis beyond 1 h reduced the subsequent proliferation of hTERT-RPE1 p53$^{WT}$ cells with a sharp threshold between 60 and

90 min, reaching a maximum after 2 h (Fig. 1d). In contrast, the hTERT-RPE1 p53$^{KO}$ cell line showed normal proliferation after mitotic delays even up to 18 h (Fig. 1d), demonstrating the cell-cycle arrest is mediated by a p53-dependent pathway. This p53-dependent reduction in proliferation of asynchronously growing cells with extended mitosis (Fig. 1a–c) or cells with spindle checkpoint delayed mitosis (Fig. 1d) was not associated with DNA damage in the arrested G1 cells in either case (Extended Data Fig. 2a–e), suggesting it has another cause. We therefore explored the idea that the detection of delays in mitosis is linked to a biochemical reaction consuming a key regulatory component of the p53 pathway, which thus acts as a timer reporting the length of mitosis (Fig. 1e). This led us to consider the role of MDM2, the p53 E3 ubiquitin ligase[12–14].

## MDM2 has the properties expected of a mitotic timer

The levels of p53 are regulated by ongoing synthesis and proteasomal destruction triggered by the E3 ubiquitin ligase MDM2. DNA damage

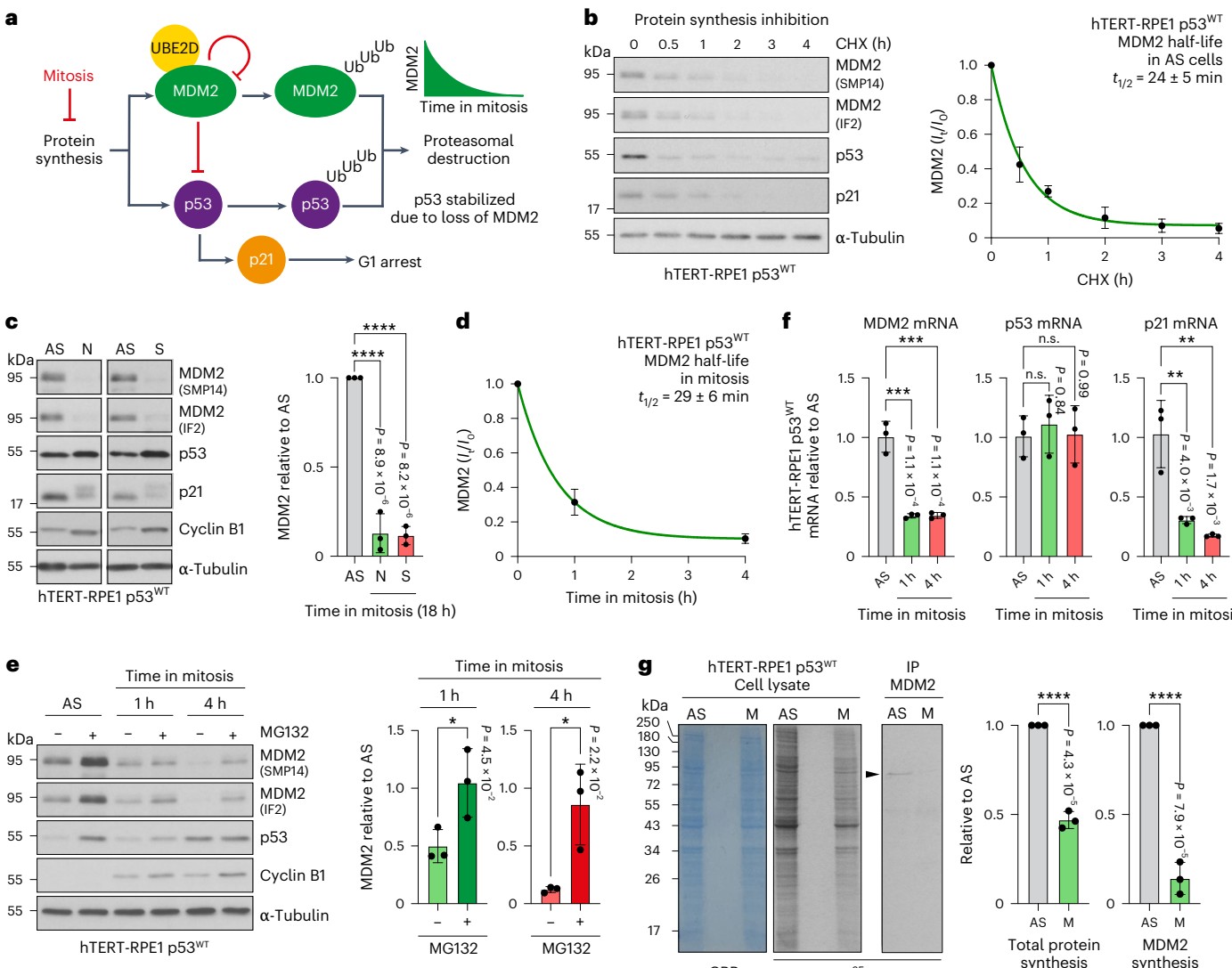

**Fig. 2 | MDM2 synthesis but not turnover is attenuated in mitosis.**
**a**, A schematic depicting MDM2 regulation of p53 through its ubiquitin (Ub) ligase activity and the role of ongoing protein synthesis and proteolysis in this pathway. **b**, MDM2 half-life was measured in asynchronous (AS) hTERT-RPE1 p53$^{WT}$ cells using western blotting by adding 50 µg ml$^{-1}$ cycloheximide (CHX) to block protein synthesis and then collecting samples up to 4 h. MDM2 half-life is indicated in the graph (mean ± s.d.; $n = 3$ independent experiments), with example western blots. MDM2 half-life in AS cells was calculated from the curve fit. **c**, The amount of MDM2 was measured by western blotting in AS or mitotic hTERT-RPE1 p53$^{WT}$ cells, arrested in mitosis for 18 h with 100 ng ml$^{-1}$ nocodazole (N) or 5 µM S-trityl L-cysteine (S) (mean ± s.d.; $n = 3$ independent experiments; $P = 8.9 \times 10^{-6}$ AS versus N, $P = 8.2 \times 10^{-6}$ AS versus S). **d**, The amount of MDM2 was measured by western blotting as a function of time relative to the starting condition ($I_t/I_0$) in AS or mitotic hTERT-RPE1 p53$^{WT}$ cells, arrested in mitosis for 1 or 4 h with 25 ng ml$^{-1}$ nocodazole (mean ± s.d.; $n = 5$ independent experiments). The mitotic MDM2 half-life was calculated from the curve fit. **e**, Same as in **d**, except the cells were treated with 20 µM MG132 to inhibit the proteasome.

The relative levels of MDM2 during 1 and 4 h mitotic delays ± proteasome inhibition are plotted (mean ± s.d.; $n = 3$ independent experiments; $P = 4.5 \times 10^{-2}$ 1 h for (−) versus (+) MG132, $P = 2.2 \times 10^{-2}$ 4 h for (−) versus (+) MG132). **f**, RT–qPCR for the level of MDM2, p53 and p21 mRNAs in AS cell cultures or arrested in mitosis for 1 or 4 h with 25 ng ml$^{-1}$ nocodazole (mean ± s.d.; $n = 3$ biological repeats, each with three technical replicates; MDM2 mRNA, $P = 1.1 \times 10^{-4}$ AS versus 1 h, $P = 1.1 \times 10^{-4}$ AS versus 4 h, p53 mRNA, $P = 0.84$ AS versus 1 h, $P = 0.99$ AS versus 4 h, p21 mRNA, $P = 4.0 \times 10^{-3}$ AS versus 1 h, $P = 1.7 \times 10^{-3}$ AS versus 4 h). **g**, Cell lysates and MDM2 immunoprecipitations (IP) from AS or mitotically arrested (M) hTERT-RPE1 p53$^{WT}$ cells labelled for 30 min with [$^{35}$S]-methionine. The relative decline in general protein synthesis and specific MDM2 synthesis between AS and M conditions are plotted on the right (mean ± s.d.; $n = 3$ independent experiments; $P = 4.3 \times 10^{-5}$ total protein synthesis AS versus M, $P = 7.9 \times 10^{-5}$ MDM2 synthesis AS versus M). IP, immunoprecipitation. The data were analysed using a one-way ANOVA test with Tukey's multiple comparisons for **c** and **f** or an unpaired two-tailed $t$-test for **e** and **g** (*$P < 0.05$, **$P < 0.01$, ***$P < 0.001$, ****$P < 0.0001$).

and other stress signals mediated by conserved signalling pathways inhibit MDM2 activity towards p53 (ref. 15). This results in p53 stabilization, an increase in the steady-state concentration of p53, and p53-dependent transcription of target genes, including MDM2, DNA repair proteins, cell-cycle regulators (such as the CDK-inhibitor p21) and, if DNA damage is not repaired, proapoptotic factors[15]. Similar to p53, MDM2 stability is tightly regulated. MDM2 has a short half-life and is subject to autoubiquitination or ubiquitination by other E3 ubiquitin

ligases[16,17]. We inferred that entry into mitosis would have important functional consequences for both MDM2 itself and MDM2 regulation of p53 at the onset of the following G1 (Fig. 2a). Upon entry into mitosis, there is a general attenuation of both transcription and translation, maintained until cells exit mitosis and re-enter G1, when transcription and protein synthesis resume[18–20]. We therefore hypothesized that MDM2 synthesis would stop or become greatly reduced following mitotic entry, whereas its turnover would continue due to ongoing

self-catalysed ubiquitination or ubiquitination by other E3 ubiquitin ligases. On this basis, we predicted that early G1 cells arising from a normal length mitosis must retain sufficient MDM2 to regulate p53 stability and thus prevent p21-induced cell-cycle arrest. However, if mitosis is prolonged and MDM2 concentration falls below this threshold, it would become limiting for p53 regulation early in the following G1, resulting in p53 stabilization and p21-dependent cell-cycle arrest. Our hypothesis is that these are general properties of the MDM2–p53 pathway, providing a mechanism to monitor the length of mitosis, independent from the cause of the delay. While elements of this hypothesis have been discussed previously[21,22], so far there is no experimental evidence in support of this proposal. We therefore set out to test these ideas and whether MDM2 has the hallmarks predicted for the mitotic timer protein.

First, we measured the stability of MDM2 in untransformed hTERT-RPE1 p53[WT] cells and a range of transformed human cancer cell lines, in both interphase and mitotic states. MDM2 expression was confirmed for all cell lines in asynchronous culture with two different antibodies (Extended Data Fig. 3a–d). HeLa cells have much lower levels of MDM2 and p53 than other cancer cell lines due to expression of the HPV E6 protein, which targets p53 for destruction, and because MDM2 is a transcriptional target of p53[23,24]. Using cycloheximide to block new protein synthesis, we found that endogenous MDM2 has a half-life of ~30 min in all of the cell lines tested, except HeLa where it is shorter due to the presence of HPV E6 (Fig. 2b and Extended Data Fig. 4). We then arrested these different cell lines in mitosis with two different antimitotic agents (nocodazole or the Eg5 kinesin inhibitor S-trityl L-cysteine) and measured the amount of MDM2 by western blotting. These approaches revealed that MDM2 falls to very low levels during an 18 h mitotic arrest in all cell lines, regardless of which agent was used to trigger the arrest (Fig. 2c and Extended Data Fig. 3c,d). In contrast, p53 was stable in a prolonged mitotic arrest in all cell lines except HeLa, consistent with the idea that its destruction requires MDM2 (Fig. 2c and Extended Data Fig. 3c). Compared with the other cell lines, HeLa cells had very low levels of p53 due to HPV E6 expression, which fell below the detection limit in the mitotic arrest samples (Extended Data Fig. 3a–d). Interestingly, the other cancer cell lines tested all showed significantly higher levels of MDM2 than hTERT-RPE1 cells (Extended Data Fig. 3a,b). To explain the decrease in MDM2 in mitosis, we next explored the roles of new protein synthesis and turnover using hTERT-RPE1 cells. MDM2 levels declined dependent on the length of time in mitosis, showing a robust decrease from 1 to 4 h of mitotic delay with an estimated half-life for MDM2 of 29 ± 6 min (Fig. 2d). Addition of the proteasome inhibitor MG132 prevented MDM2 destruction during 4 h of mitotic delay (Fig. 2e). Using quantitative PCR with reverse transcription (RT–qPCR), we then measured the levels of MDM2 messenger RNA relative to that of GAPDH in asynchronous cell cultures and cells arrested in mitosis for different lengths of time (Fig. 2f and Extended Data Fig. 5a–c). This revealed that, similar to the level of MDM2 protein, MDM2 mRNA drops rapidly following entry into mitosis within 1 h and does not recover up to 18 h of mitotic arrest (Fig. 2f and Extended Data Fig. 5c,d). Thus, we conclude that mitotic cells rapidly lose the ability to synthesize new MDM2. Similar behaviour was observed for p21 mRNA, although the protein, albeit modified in mitosis, was stable (Fig. 2c and Extended Data Fig. 5c,d). This means that new p21 synthesis in G1 following a mitotic delay would require p53-dependent transcription of the mRNA[25,26]. By contrast, p53 mRNA and protein were largely unaltered in abundance in mitotically arrested cells compared with asynchronous cells (Fig. 2c,e,f and Extended Data Fig. 5c,d). To support these conclusions, we next tested whether MDM2 protein synthesis is altered in mitosis using 30 min pulse labelling of asynchronous and mitotic cells with [35S]-methionine. Whereas the global level of protein synthesis was reduced over twofold in mitotic cells compared with an asynchronous culture, MDM2 synthesis decreased by 86% ± 7% (Fig. 2g). Thus, reduction in the level of MDM2 mRNA and global attenuation

of protein synthesis, combined with the short half-life of the MDM2 protein, can explain how the amount of MDM2 decreases in mitotic cells through a ubiquitination-dependent pathway.

## MDM2 catalyses its own destruction in mitosis

The mechanism of MDM2 turnover in mitosis is a critical component of the proposed timer function. MDM2 has been shown to undergo both self-catalysed ubiquitination, as well as ubiquitination by other E3 ubiquitin ligases[16,17]. Our data indicate that MDM2 turnover in mitosis is also likely to be via a ubiquitination-dependent pathway (Fig. 2e). We favoured a mechanism whereby MDM2 catalyses its own destruction by self-catalysed ubiquitination (Fig. 3a), since this renders the rate of MDM2 turnover less dependent on other factors including MDM2 concentration, assuming the availability of charged E2-ubiquitin conjugates is not limiting. To investigate this potential self-catalysed ubiquitination mechanism, we asked which domains of MDM2 are required for its turnover. Using cycloheximide to inhibit protein synthesis, we have mapped MDM2 turnover to the C-terminal RING domain responsible for E3 ubiquitin ligase activity towards p53 (Fig. 3b). Published structure–function data enabled us to generate specific I440A and R479A point mutants on the surface of the RING domain that prevent E2-ubiquitin binding but do not alter the structure[27]. These mutants show stabilization compared with wild-type MDM2 (Fig. 3b). In further support of the idea that E2-binding is necessary for MDM2 destruction, combined depletion of UBE2D2 and UBE2D3, the MDM2 E2 enzymes[28], resulted in a marked increase in the amount of MDM2 and stabilization of MDM2 when protein synthesis was inhibited in asynchronous cells (Fig. 3c). Moreover, in agreement with the idea that MDM2 destruction is ubiquitin-proteasome system dependent, inhibition of the proteasome stabilized MDM2 and resulted in a linear rise in MDM2 concentration over time (Fig. 3d). Combined depletion of UBE2D2 and UBE2D3 also resulted in a pronounced increase in MDM2 stability during both mitosis and interphase (Fig. 3e). UBE2D2 and UBE2D3 are also required for the activity of MDM2 towards p53, and thus, their depletion also stabilizes p53 (Fig. 3e) and triggers a G1 arrest (Fig. 3f,g).

To act as a timer, it is crucial that MDM2 is not destroyed to completion in every cell cycle regardless of the length of mitosis. It was therefore important to establish if either of the two major cell-cycle ubiquitin ligases associated with mitosis and G1, the anaphase promoting complex/cyclosome (APC/C) and Skp1–Cullin–F-box (SCF) play a role in MDM2 destruction[29]. MDM2 turnover was not altered by depletion of either the APC/C coactivator cell division cycle 20 homologue (CDC20) or both isoforms of the β-transducin-repeat containing protein (β-TrCP), a major substrate binding subunit for SCF, or by addition of the APC/C inhibitor proTAME (Extended Data Fig. 6a–d).

Together, these results provide support for the notion that MDM2 catalyses its own destruction in mitosis, and its half-life is an intrinsic property independent of the canonical cell-cycle ubiquitin ligases. However, because of the dual role of UBE2D as an E2 for both p53 and MDM2 itself, its manipulation does not allow us to fully test the role of MDM2 concentration as a key factor in the mitotic timer pathway.

## MDM2 destruction in mitosis results in G1 cell-cycle arrest

Testing the proposed mitotic timer mechanism is complicated, since removal or manipulation of MDM2 activity using standard genetic methods will cause p53 stabilization and cell-cycle arrest, independent from mitosis. Generic methods such as proteasome inhibition with MG132 are unsuitable due to the slow recovery from drug washout, leading to long delays in progression through mitosis (Extended Data Fig. 6e,f). We therefore needed to specifically target MDM2 within mitosis and uncouple its stability from the length of time spent in mitosis. To selectively modulate MDM2 stability, we used the MDM2 proteolysis-targeting chimera (PROTAC) reagent MD-224 (ref. 30), which combines a derivative of the MDM2-binding drug Nutlin-3a with a ligand for the Cereblon-Cullin 4A-RBX1 E3 ubiquitin ligase

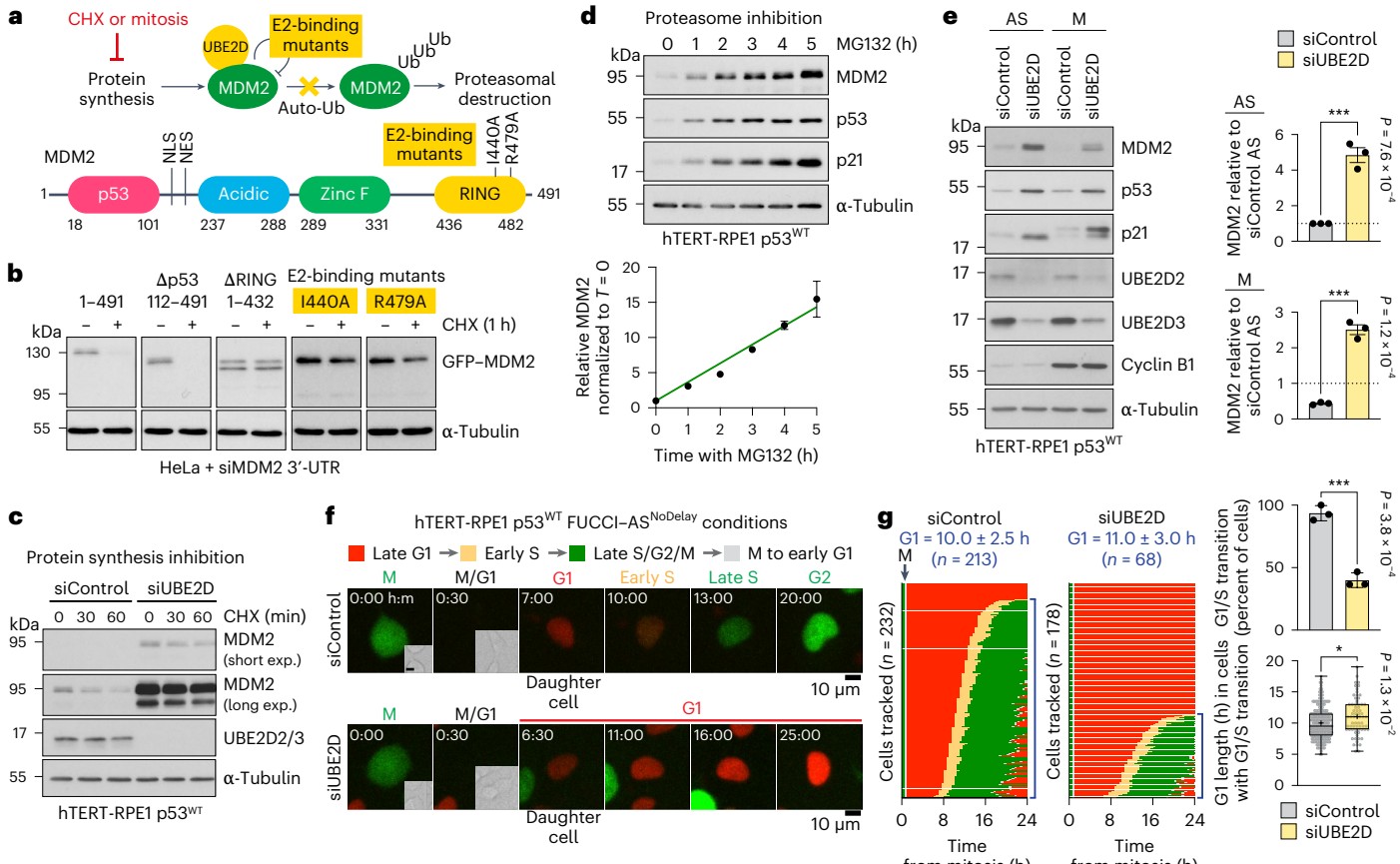

**Fig. 3 | MDM2 catalyses its own destruction in a process requiring the RING domain and UBE2D. a**, Top: a schematic depicting regulation of MDM2 by autoubiquitination. Ub, Ubiquitin. Bottom: the domain structure of MDM2 is cartooned with two E2-binding mutants, and the positions of the nuclear import (NLS) and export signals (NES) are indicated. **b**, The stability of full-length GFP–MDM2 (1–491) and deletion or point mutant constructs were tested in HeLa cells depleted of endogenous MDM2 with a 3′-untranslated region (3′-UTR) targeting siRNA, using 50 µg ml⁻¹ cycloheximide (CHX) to block protein synthesis for 1 h (+). Solvent lacking cycloheximide was used as a control (−); n = 3 independent experiments with similar results. **c**, The stability of endogenous MDM2 was tested in hTERT-RPE1 p53^WT cells, depleted of the MDM2 E2 enzymes UBE2D2/3 (siUBE2D), using 50 µg ml⁻¹ CHX up to 1 h. Short and long exposures (exp.) of Western blots are shown for MDM2. A non-targeting siRNA was used as the control; n = 3 independent experiments with similar results. **d**, The amount of MDM2 was measured by western blotting in hTERT-RPE1 p53^WT cells treated for 0–5 h with 20 µM of the proteasome inhibitor MG132. The relative MDM2 levels at each timepoint are plotted on the right (mean ± s.e.m.; n = 3 independent experiments). **e**, Endogenous MDM2 stability was measured in hTERT-RPE1 p53^WT cells depleted of the MDM2 E2 enzymes UBE2D2/3 (siUBE2D) for 30 h in asynchronous culture (AS) or after 2 h arrest in mitosis using 25 ng ml⁻¹

nocodazole (M). Non-targeting siRNA was used as a control. The relative levels of MDM2 in AS and M for each siRNA condition are plotted on the right (mean ± s.e.m.; n = 3 independent experiments; P = 7.6 × 10⁻⁴ AS siControl versus siUBE2D, P = 1.2 × 10⁻⁴ M siControl versus siUBE2D). **f**, Representative images of control or UBE2D-depleted hTERT-RPE1 p53^WT FUCCI cells exiting mitosis and entering the following cell cycle. The mitotic cells from AS culture (AS^No delay) were tracked and imaged up to 60 h post mitosis. **g**, Single cell traces of hTERT-RPE1 p53^WT FUCCI cells treated as in **f** passing from mitosis into the following cell cycle. For each condition, the number of individual daughter cells analysed are indicated. For cells that did not arrest in G1 and entered the next S phase, the mean length of G1 along with the number of cells undergoing the G1/S transition are indicated above each panel of traces (mean ± s.d.). The percentage of cells undergoing the G1/S transition is plotted in the bar graph, and the duration of G1 in those cells that entered the next S phase is depicted in a box and whiskers plot of the median, the 25th and 75th percentiles, whiskers extending to minimum and maximum values and the mean (+) for the different conditions; n = 3 independent experiments per condition; P = 3.8 × 10⁻⁴ G1/S transition siControl versus siUBE2D, P = 1.3 × 10⁻² G1 length siControl versus siUBE2D for **f** and **g**. The data were analysed using an unpaired two-tailed t-test for **e** and **g** (*P < 0.05, ***P < 0.001).

(Extended Data Fig. 7a). In asynchronous cultures of hTERT-RPE1 p53^WT cells, we observed that MD-224 caused efficient cereblon and proteasome-dependent destruction of MDM2 after 1 h and that its removal resulted in the rapid reaccumulation of MDM2 to its normal steady-state level within 1 h (Extended Data Fig. 7b–d). The latter result was important, since we needed to add MD-224 in mitosis and then wash it out before cells exit mitosis and enter G1. Confirming the anticipated functional consequences of MDM2 destruction under these conditions, we observed p53 stabilization and, after a short delay, an increase in p21 during the MD-224 washout period (Extended Data Fig. 7c). In hTERT-RPE1 p53^KO cells, MD-224 also resulted in rapid MDM2 destruction; however, the amount of MDM2 did not recover rapidly once MD-224 was washed out (Extended Data Fig. 7c), since MDM2,

similar to p21, is a transcriptional target of p53 (ref. 24). Importantly, we confirmed that MD-224 was also active in cells arrested in mitosis and promoted efficient MDM2 destruction within 1 h in both hTERT-RPE1 p53^WT and p53^KO cells (Extended Data Fig. 7e). Compared with asynchronous cell cultures where most cells are in G1, S phase or G2, the amount of MDM2 was reduced even with 1 h mitotic delay in the absence of MD-224 (Extended Data Fig. 7e). This is consistent with the results in Fig. 2 showing loss of MDM2 in prolonged mitosis. As expected, given the mode of action of MD-224, MDM2 destruction was prevented when mitotic cells were preincubated with the proteasome inhibitor MG132 (Extended Data Fig. 7f).

These results enabled us to test the idea that the stability of MDM2 is a key component of the mitotic timer. For this purpose, we combined

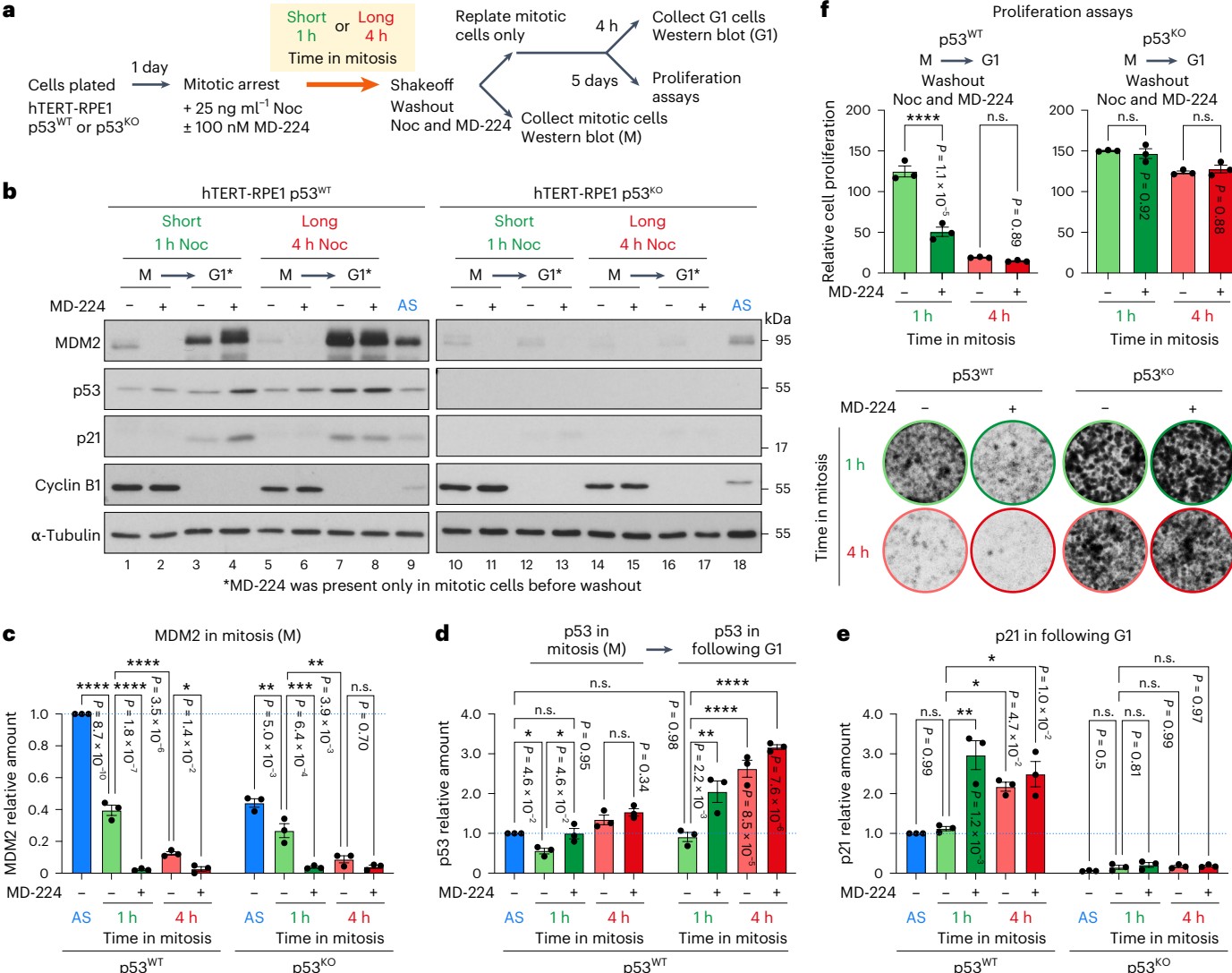

**Fig. 4 | Mitotic delays or selective MDM2 destabilization in mitosis results in p53-dependent p21 induction and reduced cell proliferation. a**, A plan of the experimental design used for **b**–**f**. **b**, Western blot of cyclin B1 positive mitotic (M) and cyclin B1 negative (G1) hTERT-RPE1 p53[WT] (lanes 1-8) or p53[KO] cells (lanes 10–17) arrested for either 1 or 4 h in mitosis with 25 ng ml⁻¹ nocodazole, in the absence (−) or presence (+) of 100 nM MD-224 for the final hour. Asynchronous hTERT-RPE1 p53[WT] (AS, lane 9) and p53[KO] cells (AS, lane 18) show the steady-state level of MDM2, p53 and p21 before entry into mitosis. **c**–**e**, Graphs calculated from **b** showing the relationship between mitotic delay and the amount of MDM2 (**c**) and p53 (**d**) in mitosis and p53 (**d**) and p21 (**e**) in the following G1 in p53[WT] and p53[KO] hTERT-RPE1 cells relative to AS p53[WT] (mean ± s.e.m.; $n = 3$ independent experiments; MDM2 in mitosis p53[WT], $P = 8.7 \times 10^{-10}$ AS versus 1 h (−)MD-224, $P = 1.7 \times 10^{-7}$ 1 h (−)MD-224 versus 1 h (+)MD-224, $P = 3.5 \times 10^{-6}$ 1 h (−)MD-224 versus 4 h (−)MD-224, $P = 1.4 \times 10^{-2}$ 4 h (−)MD-224 versus 4 h (+)MD-224, MDM2 in mitosis p53[KO] cells, p = 5.0 × 10⁻³ AS versus 1 h (−)MD-224, $P = 6.1 \times 10^{-4}$ 1 h (−)MD-224 versus 1 h (+)MD-224, $P = 3.9 \times 10^{-3}$ 1 h (−)MD-224 versus 4 h (−)MD-224, $P = 0.70$ 4 h (−)MD-224 versus 4 h (+)MD-224, p53 in mitosis, $P = 4.6 \times 10^{-2}$ AS versus 1 h (−)MD-224, $P = 4.6 \times 10^{-2}$ 1 h (−)MD-224 versus 1 h (+)MD-224, $P = 0.95$ AS versus 1 h (+)MD-224, $P = 0.34$ 4 h (−)MD-224 versus 4 h (+)MD-224, p53 in following G1, $P = 2.2 \times 10^{-3}$ 1 h (−)MD-224 versus 1 h (+)MD-224, $P = 8.5 \times 10^{-5}$ 1 h (−)MD-224 versus 4 h (−)MD-224, $P = 7.6 \times 10^{-6}$ 1 h (−)MD-224 versus 4 h (+)MD-224, $P = 0.98$ AS versus 1 h (−)MD-224, p21 in following G1 p53[WT] cells, $P = 0.99$ AS versus 1 h (−)MD-224, $P = 1.2 \times 10^{-3}$ 1 h (−)MD-224 versus 1 h (+)MD-224, $P = 4.7 \times 10^{-2}$ 1 h (−)MD-224 versus 4 h (−)MD-224, $P = 1.0 \times 10^{-2}$ 1 h (−)MD-224 versus 4 h (+)MD-224, p21 in following G1 p53[KO] cells, $P = 0.50$ AS versus 1 h (−)MD-224, $P = 0.81$ 1 h (−)MD-224 versus 1 h (+)MD-224, $P = 0.99$ 1 h (−)MD-224 versus 4 h (−)MD-224, $P = 0.97$ 1 h (−)MD-224 versus 4 h (+)MD-224). A horizontal dashed line in the graphs indicates the steady-state level of MDM2, p53 and p21, respectively, in p53[WT] cells. **f**, Cell proliferation was measured for both hTERT-RPE1 p53[WT] or p53[KO] mitotic cells treated as in **b**, using colony formation assays. Representative stained wells are shown (mean ± s.e.m.; $n = 3$ independent experiments; p53[WT] cells, $P = 1.1 \times 10^{-5}$ 1 h (−)MD-224 versus 1 h (+)MD-224, $P = 0.89$ 4 h (−)MD-224 versus 4 h (+)MD-224, p53[KO] cells, $P = 0.92$ 1 h (−)MD-224 versus 1 h (+)MD-224, $P = 0.88$ 4 h (−)MD-224 versus 4 h (+)MD-224). The data were analysed using one-way ANOVA tests with Dunnett's T3 multiple comparisons or with Tukey's multiple comparisons for **c**–**e** and **f** (*$P < 0.05$, **$P < 0.01$, ***$P < 0.001$, ****$P < 0.0001$).

a short 1 h or long 4 h spindle checkpoint arrest with MD-224 to destabilize MDM2 in hTERT-RPE1 p53[WT] and p53[KO] cells (Fig. 4a). Biochemical analysis of these cells confirmed that MDM2 was still present after a 1 h mitotic delay, albeit reduced in amount in comparison with asynchronous cells (Fig. 4b,c) and was further reduced to undetectable levels when MD-224 was added during that 1 h delay (Fig. 4b,c). After a 4 h delay, MDM2 was strongly reduced, even in the absence of MD-224,

and was further reduced to undetectable levels if MD-224 was present (Fig. 4b,c). In p53[WT] cells, p53 and p21 increased significantly in the G1 following a 1 h mitotic delay, when MDM2 was destabilized in mitosis by MD-224, to the levels seen after a 4 h mitotic delay (see Fig. 4b,d for a comparison of lanes 3–4 with lanes 7–8). Crucially, a short 1 h mitotic delay in the presence but not the absence of MD-224 triggered a cell-cycle arrest and loss of proliferation in p53[WT] cells (Fig. 4f). A long

4 h mitotic delay triggered a cell-cycle arrest and loss of proliferation in both the presence and absence of MD-224 (Fig. 4f), suggesting that after 4 h in mitosis MDM2 concentration had already fallen below the threshold needed to prevent p53 stabilization and induction of p21. Matching this behaviour, p53 was stabilized in the subsequent G1, leading to upregulation of its transcriptional targets, including MDM2 (see Fig. 4b,d for a comparison of lanes 3–4 with 7–8). This behaviour confirmed that MD-224 had been efficiently removed before G1 entry, supporting the view that its effects are due to the destabilization of MDM2 in mitosis. Confirming the dependence on p53 for the G1 arrest, induction of p21 and reduced cell proliferation were not observed with hTERT-RPE1 p53[KO] cells for any of these conditions (Fig. 4b,e,f). Note that because MDM2 is a transcriptional target of p53 it shows reduced steady-state levels in p53[KO] cells compared with p53[WT] cells (see Fig. 4b,c for a comparison of lane 9 AS p53[WT] with lane 18 AS p53[KO]). The amount of MDM2 remaining at the end of mitosis is therefore an important determinant for whether cells undergo p53-dependent arrest in the following G1.

### Defining the MDM2 threshold in mitosis for G1 arrest

Next, the cell-cycle path taken by individual cells undergoing normal length or delayed mitosis in the presence and absence of MD-224 was analysed. Single cell tracing of hTERT-RPE1 p53[WT] and p53[KO] FUCCI cells was used to follow mitotic exit, the length of G1 and entry into S phase (Extended Data Fig. 8a,b)[11]. For these experiments, mitotic cells were tracked after nocodazole washout following a short or long mitotic delay or from asynchronous cultures (no delay). The p53[WT] cells delayed in mitosis for a short period showed a similar G1 duration of 9.8 ± 3.5 h (p53[WT] Noc[Short] in Fig. 5a and Extended Data Fig. 8c) to unperturbed cells (p53[WT] AS[No delay] control in Fig. 5b and Extended Data Fig. 8c). However, even a short delay in mitosis increased the proportion of p53[WT] cells arrested in G1 for at least 20 h to 53% ± 1% (Fig. 5a,d p53[WT] Noc[Short]) from 3% ± 2% (Fig. 5b,d p53[WT] AS[No delay] control). In comparison, a long mitotic delay (p53[WT] Noc[Long]) resulted in 93% ± 1% G1 arrest, and the length of G1 in the 7% of p53[WT] cells entering S phase increased to 13.1 ± 5.5 h (Fig. 5a,d and Extended Data Fig. 8c, p53[WT] Noc[Long]). In a key test of our hypothesis, addition of 100 nM MD-224 during the short mitotic delay to p53[WT] cells resulted in 96% ± 2% arrest in G1, with correspondingly few S-phase cells after a prolonged time >20 h in G1 (Fig. 5a,d and Extended Data Fig. 8c, p53[WT] Noc[Short] + MD-224). Importantly, similar results were obtained using a CENP-E inhibitor to prevent chromosome congression (Extended Data Fig. 9a–f, p53[WT] CENP-E$_i$[Short] ± MD-224), demonstrating that the observed effect was independent of the drug used to cause mitotic delay. Shortening the Noc[Short] delay by addition

of an MPS1 inhibitor to collapse the spindle checkpoint reduced the percentage of p53[WT] cells arresting in G1 from 53% ± 1% to 29% ± 4% (Fig. 5c,d) and reduced G1 length in p53[WT] cells entering S phase to 9.6 ± 3.3 h, despite the presence of some chromosome segregation defects (Fig. 5c,d and Extended Data Fig. 8c, Noc[Short] → (+)MPS1i). In all cases the G1 arrest was robust and maintained for at least 60 h, the maximum time imaged in these experiments and was not observed in hTERT-RPE1 p53[KO] FUCCI cells for either the long mitotic arrest (Fig. 5a,d and Extended Data Fig. 8c, p53[KO] Noc[Long]) or when MD-224 was added during a short mitotic arrest (Fig. 5a,d and Extended Data Fig. 8c, p53[KO] Noc[Short] + MD-224). Similar results linking MDM2 levels in mitosis to G1 arrest were also obtained in unperturbed cells entering mitosis in asynchronous cultures without any prior cell cycle block. MDM2 destruction triggered by transient MD-224 treatment of mitotic cells in asynchronous culture resulted in an extension of G1 from 10.2 ± 3.4 h to 14.5 ± 2.6 h and increased the level of G1 arrested p53[WT] cells from 3% ± 2% to 64% ± 5% (Fig. 5b,d compare p53[WT] control with MD-224). By contrast, transient inhibition of MDM2 activity in mitosis with Nutlin-3a, an MD-224 related compound, which blocks the MDM2-p53 interaction but does not trigger MDM2 destruction, did not result in extended G1 or cell-cycle arrest in the following G1 (see Fig. 5b,d for a comparison p53[WT] control with Nutlin-3a). Reduction of the level of MDM2 in mitosis and not simply inhibition of its activity is, therefore, critical for G1 arrest. Importantly, transient treatment of cells in S phase and G2 with MD-224 did not trigger arrest in G1 phase of the following cell cycle, and all cells entered S phase, although we did observe a lengthened G1 (Fig. 5e). This is consistent with our data showing rapid recovery of MDM2 following washout of MD-224 in asynchronous cell cultures (Extended Data Fig. 7c). Taken together, these results show that the proportion of cells undergoing a G1 arrest and length of G1 correlate with the increased length of time in mitosis or reduced level of MDM2.

To understand the consequences of delays in mitosis and accompanying reduced levels of MDM2 for early events in the following G1, we then measured p21 levels in single hTERT-RPE1 p53[WT] p21–GFP cells[31]. We observed that after a short 1 h mitotic arrest the level of p21 was slightly increased in early G1 cells, whereas after a longer 4 h delay or when 100 nM MD-224 was added during the short mitotic delay and then removed, p21 levels rose sharply in early G1, plateaued and were then maintained into late G1 (Fig. 5f). To determine the precise threshold below which MDM2 has to be reduced in mitosis to trigger p21 induction in G1, we titrated MD-224 in the hTERT-RPE1 p53[WT] FUCCI cell line and measured the amount of MDM2 by western blotting, and both nuclear levels of p21 and cell-cycle arrest by microscopy. This revealed a sharp dose-response relationship, where G1 arrest is closely

**Fig. 5 | Defining the MDM2 threshold in mitosis for p21 induction and robust cell-cycle arrest in G1. a**, Single cell traces of hTERT-RPE1 p53[WT] and p53[KO] FUCCI cells passing from mitosis into the following cell cycle after arrest in mitosis (M) for either 30 min (Noc[Short]) or 4 h (Noc[Long]) with 25 ng ml⁻¹ nocodazole, in the absence or presence of 100 nM MD-224 for 30 min before washout. The mitotic cells were tracked after washout and imaged up to 60 h post mitosis (*n* = 3 independent experiments per condition). **b**, Mitotic hTERT-RPE1 p53[WT] FUCCI cells in asynchronous culture (AS[No delay]) were treated for 30 min with DMSO control, MD-224 or Nutlin-3a, the drugs washed out and the cells then tracked into the following cell cycle (*n* = 3 independent experiments per condition). **c**, MPS1 inhibitor was used to override the Noc[Short] spindle assembly checkpoint arrest. Single cell traces of hTERT-RPE1 p53[WT] FUCCI cells in the Noc[Short] + MPS1i condition passing from mitosis into the following cell cycle (*n* = 3 independent experiments). **a**–**c**, For each condition, the number of individual daughter cells analysed are indicated. For cells that did not arrest in G1 and entered the next S phase, the mean length of G1 along with the number of cells undergoing the G1/S transition are indicated above each panel of traces (mean ± s.d.). The dotted line in each panel indicates the mean length of G1. Box and whiskers plots for G1 length show the median, the 25th and 75th percentiles, whiskers extending to minimum and maximum values, and the mean (+). **d**, The percentage of cells arrested in G1 is plotted (mean ± s.d.; *n* = 3 independent experiments

per condition). **e**, Single cell traces of hTERT-RPE1 p53[WT] FUCCI cells in either G1/S or S/G2 in asynchronous cultures were treated with MD-224 for 30 min as in the Noc[Short] condition, washed to remove the drug, then tracked into the following cell cycle. The proportion of cells arresting in G1 and entering the next S phase are shown (*n* = 3 independent experiments; *P* = 6.2 × 10⁻⁹ p53[WT] Noc[Short] versus p53[KO] Noc[Short], *P* = 6.5 × 10⁻⁷ p53[WT] Noc[Short] versus p53[WT] Noc[Long], *P* = 2.5 × 10⁻¹³ p53[WT] Noc[Long] versus p53[KO] Noc[Long], *P* = 2.1 × 10⁻⁷ p53[WT] Noc[Short] versus p53[WT] Noc[Short] + MD-224, *P* = 1.1 × 10⁻¹³ p53[WT] Noc[Short] + MD-224 versus p53[KO] Noc[Short] + MD-224, *P* = 7.8 × 10⁻⁴ p53[WT] Noc[Short] versus p53[WT] Noc[Short] + MPS1i, *P* = 3.4 × 10⁻¹⁰ control versus (+)MD-224, *P* = 0.99 control versus (+)Nutlin-3a). **f**, Levels of p21 in single hTERT-RPE1 p53[WT] p21–GFP cells were followed post mitosis in new G1 cells for the Noc[Long] (*n* = 33 cells), Noc[Short] (*n* = 39 cells) and Noc[Short] + MD-224 (*n* = 36 cells) conditions (mean ± s.e.m.; *n* = 3 independent experiments per condition). **g**, Titration of MD-224 to reveal the threshold concentration required for G1 arrest (black line) of hTERT-RPE1 p53[WT] FUCCI cells. MDM2 levels were determined by western blotting (mean ± s.e.m.; *n* = 4 independent experiments) and p21 levels (mean, red) and proportion of cells arrested in G1 (black) by immunofluorescence microscopy (mean ± s.e.m.; *n* = 3 independent experiments). The data were analysed using a one-way ANOVA test with Tukey's multiple comparisons for **d** (***P* < 0.001, ****P* < 0.0001).

correlated with the increasing level of p21 and decreasing amount of MDM2 (Fig. 5g). The 100 nM concentration of MD-224 used for other experiments sits at the top of this sigmoidal curve, explaining its potency. Collectively, our data therefore reveal a defined threshold for MDM2 in mitosis, below which p53 stabilization and, hence, p21 induction in G1 cells leads to cell-cycle arrest.

**MDM2 overexpression increases the time in mitosis threshold**

We next asked if increasing the level of MDM2 by overexpression can overcome the mitotic timer response by using hTERT-RPE1 p53[WT] cells with a stably integrated GFP–MDM2 construct (GFP–MDM2[OE]). Western blotting showed that GFP–MDM2[OE] cells express around fivefold

more GFP–MDM2 than endogenous MDM2 in hTERT-RPE1 p53[WT] cells (Fig. 6a,b and Extended Data Fig. 10a). Both proteins have a similar half-life of 22–24 min (Extended Data Fig. 10b); however, due to its increased level in the GFP–MDM2[OE] cells, MDM2 will, therefore, take longer to drop below the critical threshold for p21 induction. Supporting this idea, the threshold for nuclear p21 induction, determined by MD-224 titration, was increased from $9.5 \pm 1.7$ nM in hTERT-RPE1 p53[WT] to $26.9 \pm 3.7$ nM MD-224 in GFP–MDM2[OE] cells (Fig. 6b), with comparable results for the total p21 induction threshold obtained by western blotting (Extended Data Fig. 10c). The response of these cells to 1 or 4 h mitotic delay was then compared in mitotic timer and cell proliferation assays. Compared with control hTERT-RPE1 p53[WT] cells,

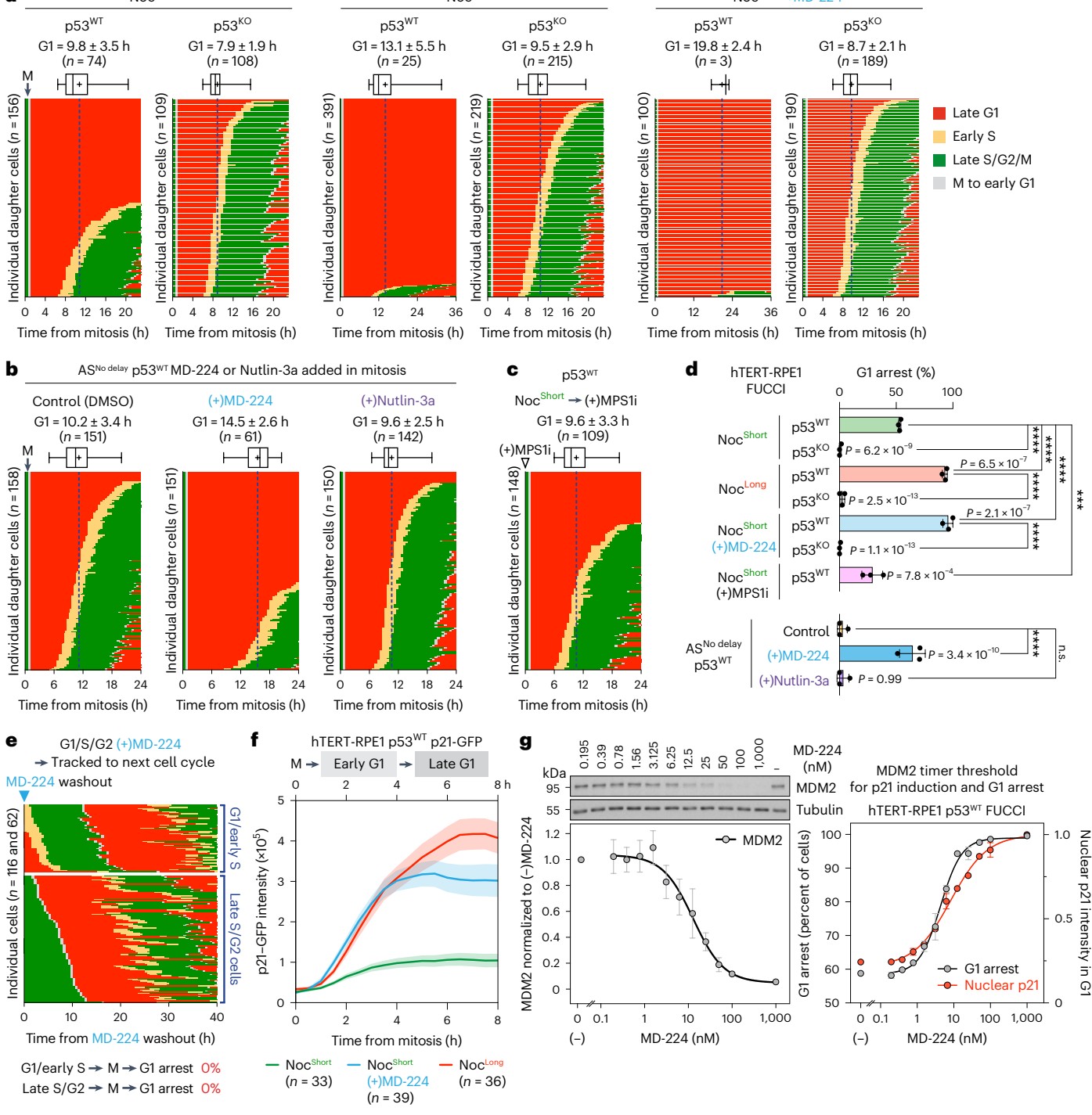

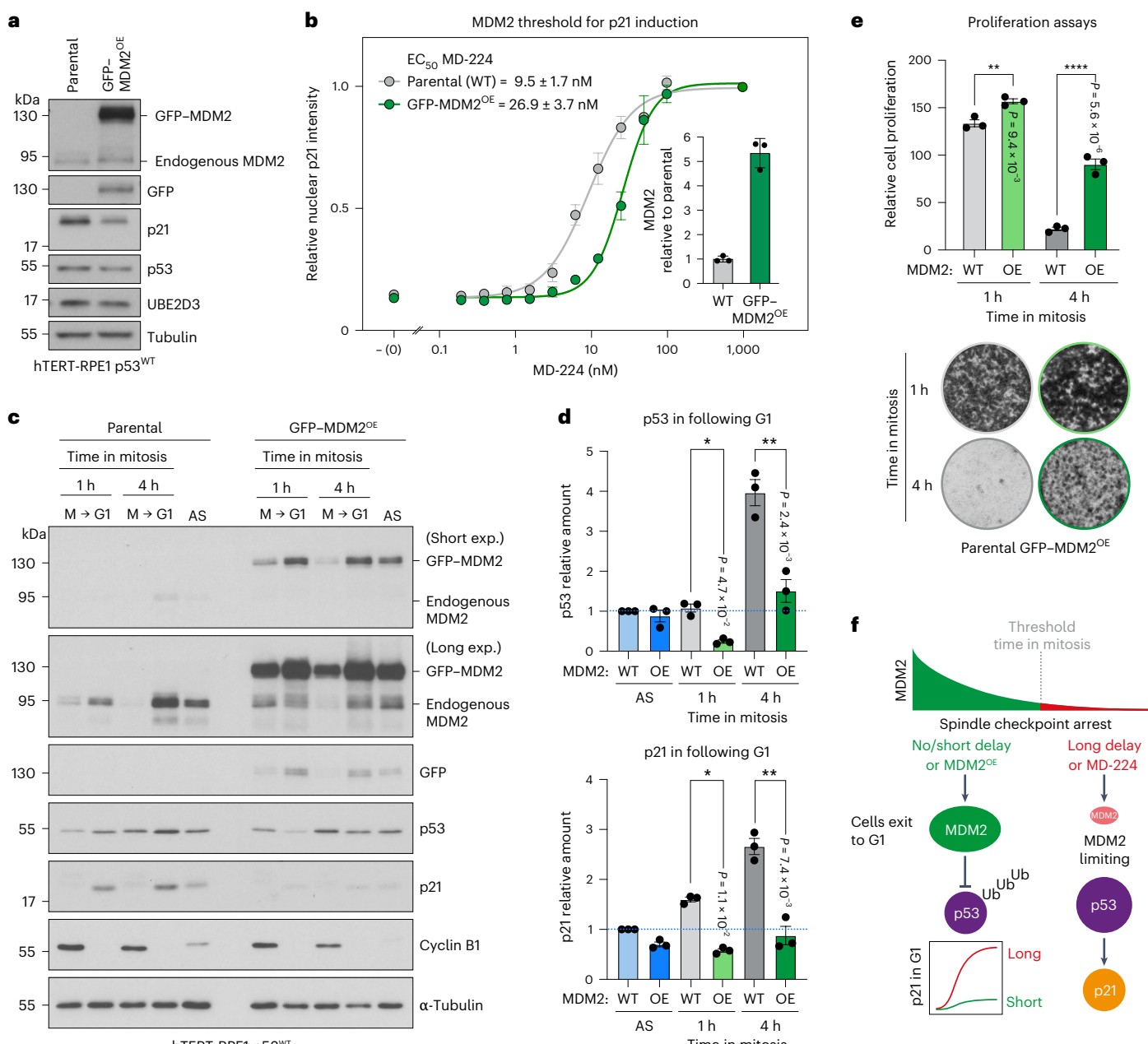

**Fig. 6 | MDM2 overexpression attenuates the mitotic timer response.**
**a**, Western blot of MDM2 levels in hTERT-RPE1 p53$^{WT}$ (parental) and GFP−MDM2$^{OE}$ cells. Three independent experiments showed similar results. **b**, A titration of MD-224 to reveal the threshold concentration required for p21 induction. Nuclear p21 levels were measured by immunofluorescence microscopy ($n = 1,034–6,124$ cells were analysed for each MD-224 concentration in hTERT-RPE1 p53$^{WT}$ (grey line, circles) and GFP−MDM2$^{OE}$ cells (green line, circles); mean ± s.e.m.; $n = 3$ independent experiments). The relative level of MDM2 in GFP−MDM2$^{OE}$ cells compared with parental cells is shown in the bar graph calculated from Extended Data Fig. 10a (mean ± s.d.; $n = 3$ independent experiments). **c**, Western blot of cyclin B1 positive mitotic (M) and cyclin B1 negative (G1) hTERT-RPE1 p53$^{WT}$ (parental) or GFP−MDM2$^{OE}$ cells arrested for either 1 or 4 h in mitosis with 25 ng ml$^{-1}$ nocodazole. Asynchronous hTERT-RPE1 p53$^{WT}$ cells (AS) were used as a normalization control to allow direct comparison of the steady-state levels of p53 and p21. Short and long exposures (exp.) of MDM2 western blots are shown to enable visualisation of endogenous MDM2. **d**, Graphs calculated from **c** showing the relationship between mitotic delay and the amount of p53 and p21 in the

following G1 in parental (WT) and GFP−MDM2$^{OE}$ (OE) hTERT-RPE1 p53$^{WT}$ cells (mean ± s.e.m.; $n = 3$ independent experiments; p53 in following G1, $P = 4.7 \times 10^{-2}$ WT 1 h versus OE 1 h, $P = 2.4 \times 10^{-3}$ WT 4 h versus OE 4 h, p21 in following G1, $P = 1.1 \times 10^{-2}$ WT 1 h versus OE 1 h, $P = 7.4 \times 10^{-3}$ WT 4 h versus OE 4 h). **e**, hTERT-RPE1 p53$^{WT}$ parental (WT) or GFP−MDM2$^{OE}$ (OE) cells were arrested in mitosis for 1 or 4 h with 25 ng ml$^{-1}$ nocodazole. The mitotic cells were collected and washed out from nocodazole, and 5,000 cells were plated per well. After 5 days, cell proliferation was measured by crystal violet staining and representative stained wells are shown (mean ± s.e.m.; $n = 3$ independent experiments; $P = 1.0 \times 10^{-2}$ WT 1 h versus OE 1 h, $P = 5.6 \times 10^{-6}$ WT 4 h versus OE 4 h). **f**, A schematic depicting the decay in MDM2 levels during mitosis. If MDM2 concentration drops below a threshold in mitosis, MDM2 becomes limiting for p53 regulation in the ensuing G1, leading to p53 stabilization, p21 induction and G1 cell-cycle arrest. Ubiquitin (Ub) addition to p53 by MDM2 fails to occur after long delays in mitosis or treatment with MD-224. The data were analysed using a one-way ANOVA test with Tukey's multiple comparisons for **d** and **e** (*$P < 0.05$, **$P < 0.01$, ****$P < 0.0001$).

GFP–MDM2[OE] cells failed to stabilize p53 or induce p21 in G1 following a 4 h delay in mitosis (Fig. 6c,d). Furthermore, GFP–MDM2[OE] cells did not exhibit the characteristic arrest of cell proliferation observed for the hTERT-RPE1 p53[WT] cells after a 4 h delay in mitosis (Fig. 6e). Increasing the steady-state level of MDM2, therefore, suppresses the mitotic timer response, lending further support to the idea that MDM2 concentration and, hence, p53 stabilization is the crucial determinant reporting when the length of mitosis exceeds a crucial time threshold.

## Discussion

These findings support the conclusion that MDM2, due to its short half-life, is a key timer component in the mechanism that triggers a robust cell-cycle arrest in the G1 following a prolonged delay in mitosis[21,22]. This is reminiscent of the role played by the antiapoptotic protein Mcl-1 in the regulation of apoptosis after extended mitosis. In that case, slow APC/C-dependent destruction of Mcl-1 acts as a timer for apoptosis[32,33]. By contrast, MDM2's timer properties arise through a self-catalysed ubiquitination and proteasomal destruction mechanism and the attenuation of protein synthesis in mitosis. When MDM2 drops below a threshold concentration in mitosis, it becomes limiting for p53 destruction, leading to p53 stabilization and p21 induction in new G1 cells (Fig. 6f). Other studies suggest that a PLK1-regulated stopwatch complex of p53BP1 and USP28, the MDM2 counteracting p53-deubiquitinating enzyme, is important for G1 arrest following long mitotic delays caused by loss of centrosomes or microtubule depoly-merising drugs[7–10,34,35]. This complex shows the opposite behaviour to MDM2, increasing during extended mitosis[34,35], which, given the role of USP28 as an antagonist of MDM2, might further stabilize p53 in new G1 cells. However, as G1 arrest behaviour can be uncoupled from time in mitosis using the MDM2 PROTAC MD-224, we conclude that MDM2 concentration is the initial limiting component in the mitotic timer pathway. Supporting this view, MDM2 becomes limiting after 60–90 min of mitosis, with a sharp threshold matching the G1 arrest behaviour and induction of the CDK-inhibitor p21. The PLK1-regulated p53BP1 and USP28 stopwatch complex appears to accumulate more slowly becoming detectable from 2–4 h arrest in mitosis[35], suggesting this may reinforce the loss of MDM2 during longer mitotic delays.

Our observations help provide an explanation for p53-dependent, yet DNA damage signalling-independent, cell-cycle arrest in G1 following delays in mitosis[1–10]. Loss-of-function mutations or deletion of p53 are among the most common genetic changes associated with aneuploid cancers, and we propose this is in part due to the central role of MDM2 in the mitotic timer pathway. Highly aneuploid tumour cells often show prolonged mitosis due to extended spindle assembly checkpoint activation caused by mitotic defects[1–10]. As long as p53[WT] functionality is present, the mechanism we have described here will channel these cells into a prolonged cell-cycle arrest in the following G1. Conversely, loss of this protective response enables aneuploid cells or cells harbouring other changes that delay progression through mitosis to evade G1 arrest and continue proliferating. These findings shed new light on to the functions of MDM2 and have potentially important consequences for both our understanding of how aneuploidy is detected in normal cells and how failure of this process facilitates genome instability and unchecked proliferation in cancers.

## Online content

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

## Methods

### Reagents and antibodies

Laboratory chemicals were obtained from Merck, Sigma-Aldrich and Thermo Fisher Scientific. Commercially available polyclonal antibodies or monoclonal antibodies used for western blotting, immunofluorescence and immunoprecipitation are listed in Supplementary Methods Table 1.

### Inhibitors and drug compounds

Commercially sourced inhibitors and drugs used in this study and the solvents used to reconstitute them are listed in Supplementary Methods Table 2. Experiment-specific timings and concentrations are given in the figure legends.

### MDM2 expression plasmid construction

Mammalian eGFP–MDM2 expression constructs were made in pEF5/FRT or pcDNA5/FRT plasmids harbouring eGFP followed by a 25 amino acid linker YK(GSSS)$_5$RIP 5′. The full-length MDM2 coding sequence was isolated from Addgene plasmid no. 16233 (pcDNA3 FRT MDM2) by PCR using the *Thermococcus kodakaraensis* (KOD) hot-start polymerase kit (Merck no. 71086) and primers that included a 5′ BamHI restriction site and a 3′ XhoI restriction site. Following BamHI (NEB no. R3136)-XhoI (NEB no. R0146) digest of the MDM2 PCR product and destination vector, ligations were performed with T4 ligase (NEB no. M0202). The ligated plasmids were transformed into XL-1 blue cells (Agilent Technologies no. 200249). I440A and R479A MDM2 mutants were generated by site-directed mutagenesis using KOD polymerase. The MDM2 truncated fragments were generated by PCR using Addgene plasmid no. 16233 (pcDNA3 FRT MDM2) as a template and cloned into pEF5 FRT eGFP with BamHI-XhoI. The primers used to generate constructs are listed in Supplementary Methods Table 3. The mammalian expression eGFP–MDM2 overexpression plasmid was generated by subcloning the eGFP–MDM2 insert into a pMK232 vector (Addgene no. 72834) digested with NdeI (NEB no. R0111S)-XhoI (NEB no. R0146).

### Cell lines and cell culture

All cell lines are validated stocks purchased from ATCC. The hTERT-RPE1 p53^WT and p53^KO cell lines were authenticated by STR profiling (ATCC). HeLa, HCT116 and hTERT-RPE1 FUCCI cell lines were authenticated by STR profiling (NorthGene). All hTERT-RPE1 cell lines (no. CRL-4000) were cultured in DMEM-F12 Ham medium (Sigma no. D6421), supplemented with GlutaMAX (Gibco no. 35050087) and 10% (v/v) foetal bovine serum (FBS, Sigma-Aldrich no. F9665). HeLa (no. CRL-2.2), A375 (no. CRL-1619), U2OS (no. HTB-96) and HEK293T (no. CRL-3216) cells were cultured in DMEM buffer containing GlutaMAX (Gibco no. 10569010) and 10% (v/v) FBS. HCT116 cells (no. CCL-247) were cultured in McCoy's 5A medium (Gibco no. 16600082), supplemented with 10 mM sodium pyruvate (Thermo Fisher no. 11360088) and 10% (v/v) FBS. The hTERT-RPE1 p53^WT p21–GFP cell line has been described previously[31]. For routine passaging of cells, the cells were washed in PBS and incubated with TrypLE Express Enzyme cell dissociation reagent (Gibco no. 12605036) for 5 min at 37 °C, before resuspending detached cells in full medium for passage. All cell lines were maintained at 37 °C, humidified 5% CO$_2$ in a cell culture incubator (Thermo, HERAcell, no. 51013568). Mycoplasma negative status of cell lines was confirmed using the EZ-PCR Mycoplasma Test Kit with internal control (K1-0210, Geneflow). The cell lines used in our studies are not on the list as commonly misidentified lines.

### Transient transfection of cells

The cells were seeded at a density of 50,000 cells per well in six-well plates. The next day, transfection mixes were prepared in sterile DNase-free Eppendorf tubes (Eppendorf no. 0030108035) consisting of 100 µl Opti-MEM (Gibco no. 11058021), 3 µl Mirus TransIT-LT1 transfection reagent (Mirus no. MIR2300) and up to 1 µg of plasmid DNA.

If less than 1 µg plasmid DNA was required, pBlueScript II (Agilent) was used to bring the total amount of DNA to 1 µg. The transfection mixes were vortexed briefly for 20 s and incubated at room temperature for 30 min before adding dropwise to cells. The cells were then left to grow for the times indicated in the figure legends. To generate the eGFP–MDM2 overexpressing cell line, 800 ng of the pMK232 eGFP–MDM2 plasmid was transfected into p53^WT hTERT-RPE1 cells, as described above. A total of 48 h post-transfection, the cells were selected with puromycin and diluted to one cell per well of a 96-well plate. The viable colonies were screened for GFP–MDM2 overexpression.

### CRISPR/Cas9 gene editing of hTERT-RPE1 cells

First, hTERT-RPE1 cells (ATCC, no. CRL-4000) were edited to inactivate the puromycin resistance cassette through delivery of two pX461 vectors (Addgene no. 48140) containing the appropriate guide RNA (gRNA) sequences (CGCTCAACTCGGCCATGCGC; GCAACAGATGGAAGGC-CTCC) using the Amaxa nucleofector kit V (Lonza no. VCA-1003). The monoclonal lines were screened for puromycin sensitivity, growth rate and cilium formation. We refer to this cell line as hTERT-RPE1 p53^WT. Subsequently, to generate p53^KO hTERT-RPE1 cells, the pX459 vector containing guide RNA against p53 (CCATTGTTCAATATCGTCCG) was delivered through electroporation. After brief selection in puromycin, the monoclonal lines were screened for Nutlin-3a resistance, followed by extensive quality controls including immunoblotting and amplification of the targeted genomic region by PCR. The genomic DNA was isolated from clones using QuickExtract DNA isolation solution (Lucigen no. QE09050). The PCRs were performed with KOD hot-start polymerase (Merck no. 71086), using primers flanking the gRNA recognition site. The PCR products were ligated into pSC-A vectors using a Blunt-end PCR cloning kit (Agilent no. 240207). The ligated plasmids were transformed into XL-1 blue competent cells, and around ten colonies were miniprepped (Qiagen no. 27107) and sequenced using Sanger DNA sequencing (Source Genomics). The p53^KO cell lines were STR profiled using the ATCC cell line authentication service (ATCC) and validated as hTERT-RPE1. We refer to this cell line as hTERT-RPE1 p53^KO.

### hTERT-RPE1 FUCCI cell line generation

FUCCI reporter constructs were introduced into hTERT-RPE1 p53^WT and p53^KO cells by lentiviral infection. Briefly, pBOB-EF1-FastFUCCI-Puro (Addgene, no. 86849), pMD2.G (Addgene no. 12259) and psPAX2 (Addgene no. 12260) (gifts from Didier Trono) were used for lentiviral-based packaging in HEK293T cells for 2 days with a media change after 24 h. The resulting lentiviral supernatant was used to infect hTERT-RPE1 p53^WT and p53^KO for 24 h. After infection, antibiotic-resistant clones were selected by puromycin and then expanded in non-selective medium to be screened by fluorescence microscopy for successful integration of the FUCCI sensor.

### RNA interference

The cells were seeded at a density of 30,000–50,000 cells per well in six-well plates. The next day, transfection mixes were prepared in sterile DNase-free Eppendorf tubes consisting of 100 µl Opti-MEM (Gibco no. 11058021), and either 3 µl Mirus TransIT-X2 transfection reagent (Mirus no. MIR6000) for hTERT-RPE1 cells or 3 µl Oligofectamine (Thermo no. 12252011) for HeLa cells. The solutions were mixed and incubated for 5 min at room temperature, then 60 pM short interfering RNA (siRNA) was added, followed by incubation at room temperature for 30 min, before adding dropwise to cells. The cells were left to grow for the times indicated in the figure legends. The siRNAs used in this study are listed in Supplementary Methods Table 4.

### RT–qPCR procedure

The cells were seeded for 70–80% confluency 24 h before lysis. The cells were treated as indicated in the figure legends and washed in PBS, and RNA was extracted using the RNeasy Micro Kit (Qiagen no. 74004)

following the manufacturer's instructions. RNA purity and yield were assessed on a Nanodrop (Thermo Fisher), and 1–1.5 µg of RNA was used a template to make cDNA with the iScript cDNA synthesis kit (Bio-Rad no. 1708890), as per the manufacturer's instructions. cDNA was diluted 1:10 with nuclease-free water. For RT–qPCR, 4 µl of the diluted cDNA was added to a Master mix containing 4 µl SsoAdvanced Universal SYBR Green Supermix (Bio-Rad no. 1725270) and 0.5 µl of a 10 µM stock of each forward and reverse primer (Supplementary Methods Table 5) per reaction in a 384-well plate format (Bio-Rad no. HSP3805). The plates were sealed (Thermo Fisher no. AB0558) and briefly spun down at 500$g$ for 30 s and RT–qPCR was performed using a CFX384 Touch Real-Time PCR Detection system (Bio-Rad no. 1855484). To confirm amplification of the intended targets, the PCR products were loaded on to 1% (w/v) agarose gels stained with Midori Green Advance DNA stain (Geneflow no. S6-0022) alongside 100 bp ladder (NEB no. N3231S), before submitting for DNA Sanger sequencing (Source Genomics).

### Mitotic timer assays

For mitotic timer assays, hTERT-RPE1 p53$^{WT}$ or p53$^{KO}$ cells were seeded into 10 cm dishes (five dishes per condition) and left to grow until 80% confluency, typically the next day. The cells were treated with 25 ng ml$^{-1}$ nocodazole for 45 min at 37 °C. The mitotic cells were isolated by shake-off and washed three times in prewarmed 37 °C growth medium; the cells were pelleted at 200$g$ for 3 min at 37 °C in between washes. A mitotic sample was prepared by lysis of half of the cells. The remaining cells were replated for 4 h at 37 °C to allow entry into G1. For longer mitotic arrests, following mitotic shake-off, the cells were pelleted and resuspended in growth medium containing 25 ng ml$^{-1}$ nocodazole and replated into 6 cm dishes for the times indicated in the figures. Where indicated, MD-224 or dimethylsulfoxide (DMSO) (Sigma no. D8418) vehicle control was added in the last 45 min of mitosis, before the nocodazole washout procedure. The asynchronous cell cultures were used as controls.

### Cell lysis for western blots

To prepare lysates, the cells were washed in ice-cold PBS and then extracted in ice-cold lysis buffer for 10 min (20 mM Tris–HCl (Sigma-Aldrich no. T4661) pH 7.4, 150 mM NaCl (Sigma-Aldrich no. S3014), 1% (v/v) IGEPAL (Sigma-Aldrich no. S3014), 0.1% (w/v) sodium deoxycholate (Sigma-Aldrich no. D6750), 40 mM sodium β-glycerophosphate (Sigma-Aldrich no. G9422), 10 mM NaF (Sigma-Aldrich no. 201154), 0.3 mM sodium orthovanadate (Sigma-Aldrich no. S6508), 100 nM okadaic acid (Enzo Life Sciences no. ALX-350-011-M001), 200 nM microcystin-LR (Enzo Life Sciences no. ALX-350-012-M001), 1 mM dithiothreitol (DTT) (Sigma-Aldrich no. 11583786001), protease inhibitor cocktail (Roche, cOmplete, EDTA-free Protease Inhibitor Cocktail, no. 11873580001), and 1× phosphatase inhibitor cocktail (Sigma no. P0044) in water). The lysates were collected in 1.5 ml Eppendorf tubes (Eppendorf no. 10451043) and then clarified by centrifugation at 14,000 r.p.m. for 15 min at 4 °C. The protein concentrations were measured by Bradford assay using Protein Assay Dye Reagent Concentrate (Bio-Rad no. 5000006). The samples were normalized to 0.5–1 mg ml$^{-1}$. The sample buffer (0.1875 mM Tris–HCl (Sigma-Aldrich no. T4661) pH 6.8, 1% (w/v) SDS buffer (Sigma-Aldrich no. 75746), 10% (v/v) glycerol (Sigma-Aldrich no. G9012), 0.05% (w/v) bromophenol blue (Sigma-Aldrich no. B5525) and 10% (v/v) β-mercaptoethanol (Sigma-Aldrich no. M3148) in water) was added, and the samples were boiled at 95 °C for 10 min before progressing to SDS–polyacrylamide gel electrophoresis (PAGE) and western blotting.

### SDS–PAGE and western blotting

The proteins were separated by SDS–PAGE (Bio-Rad, Mini-PROTEAN Tetra cell, no. 1658000) and transferred onto nitrocellulose membranes (Bio-Rad nos. 1704158 and 1704159) using a Trans-blot Turbo system (Bio-Rad no. 1704150). After transfer, the blots were blocked in 5% (w/v) milk powder in PBS (PanReac AppliChem no. A0965,9100)-TWEEN 20 (Sigma-Aldrich no. P1379) (PBS-T), before incubation with primary antibodies (Supplementary Methods Table 1) diluted in 5% (w/v) milk in PBS-T for 1 h at room temperature or at 4 °C overnight. The blots were then washed in PBS-T before incubating with the secondary HRP-conjugated antibodies (Supplementary Methods Table 1) for 1 h at room temperature, before washing again in PBS-T. The signals were revealed using ECL (GE Healthcare no. RPN2106) and visualized on X-ray films (GE Healthcare no. GE28-9068-37), developed on an OPTIMAX 2010 X-ray Film Processor (PROTEC-Med).

### Immunoprecipitation of MDM2, p53 and p21

The pellets of $1 \times 10^6$ asynchronous or mitotic cells were lysed in 1 ml of lysis buffer (50 mM HEPES buffer pH 7.5, 100 mM NaCl, 0.5% (v/v) Triton X-100, 1 mM DTT, 1:250 Protease Inhibitor Cocktail (Merck no. P8340-5ML), 1:1,000 Phosphatase Inhibitor cocktail 3 (Merck no. P0044-5ML), 100 nM okadaic acid (Enzo LifeSciences no. ALX-350-011-M001)) on ice for 30 min, and the lysates were then clarified at 14,000 r.p.m. for 30 min at 4 °C. The asynchronous cells were detached by incubating cells in 1 mM EDTA in PBS before pelleting and lysis. The mitotic cells were isolated by shake-off after 100 ng/ml nocodazole treatment for 18 h. A 5% (v/v) aliquot of cell lysate was mixed with sample buffer as input for immunoblotting. The proteins of interest were isolated from the remaining cell lysate by immunoprecipitation with 1-2.5 µg of antibodies and 20 µl of protein G–Sepharose (Merck, GE17-0618-01) at 4 °C for 2 h. The antibody-bound beads were then washed three times with lysis buffer at 4 °C, resuspended in sample buffer and boiled at 65 °C to elute. IP and input samples were analysed by immunoblotting as described above.

### [$^{35}S$]-methionine labelling and immunoprecipitation of MDM2

The pellets of $2.5 \times 10^6$ asynchronous and mitotic cells, isolated as described in the immunoprecipitation section above, were resuspended in 1 ml of prewarmed labelling medium (Met/Cys/Gln-free DMEM (Thermo, 21013024), containing 0.2 mM L-cysteine, 2 mM GlutaMax (Thermo no. 35050061), 2.5 mM HEPES (Gibco no. 15630-056), 1 mM sodium pyruvate (Thermo no. 11360070) and 10% (v/v) dialysed FBS) with or without 100 ng ml$^{-1}$ nocodazole. A total of 200 µCi ml$^{-1}$ [$^{35}S$]-methionine was added into the cell suspensions for 30 min at 37 °C and 5% CO$_2$. Following incubation, the cells were washed in PBS and pelleted at 200$g$ for 5 min at 37 °C before lysing in 1.25 ml of lysis buffer (50 mM HEPES pH 7.5, 100 mM NaCl, 0.5% (v/v) Triton X-100, 1 mM DTT, 1:250 Protease Inhibitor Cocktail (Merck no. P8340-5ML), 1:1,000 Phosphatase Inhibitor cocktail 3 (Merck no. P0044-5ML), 100 nM okadaic acid (Enzo LifeSciences no. ALX-350-011-M001)). The lysates were incubated on ice for 30 min and then clarified at 14,000 r.p.m. for 30 min at 4 °C. An 8% (v/v) aliquot of each cell lysate was mixed with sample buffer to obtain input samples. The proteins of interest were isolated from the remaining cell lysate by immunoprecipitation with 1–2.5 µg of antibodies as described in the figures. The proteins were separated by SDS–PAGE, and the gels were fixed and stained using InstantBlue Coomassie Protein Stain (ISB1L) (Abcam no. ab119211). The gels were destained with water and then dried in a gel dryer (Bio-Rad, model no. 583). Detection of the incorporated label was performed by autoradiography, by exposing gels to X-ray films (GE Healthcare no. GE28-9068-37) for 2–7 days at −80 °C. The autoradiographs were developed using an OPTIMAX 2010 X-ray Film Processor (PROTEC-Med).

### Cell proliferation assays

To test for proliferation, hTERT-RPE1 p53$^{WT}$ or p53$^{KO}$ cells were treated as described in the mitotic timer assays methods section. Following nocodazole washout, the mitotic cells were counted using a Neubauer chamber and were seeded at 5,000 cells per well in six-well plates. The cells were cultured at 37 °C for 5 days. Following incubation, the cells

were washed in PBS and stained in 5% (w/v) crystal violet (Sigma-Aldrich no. C0775) in (25% (v/v) methanol (VWR no. 20847.307) for 30 min at room temperature. The cells were washed in water and left to dry before imaging on a Bio-Rad Gel Doc XR+ imaging system (Bio-Rad no. 1708195EDU) using Image Lab 6 software (Bio-Rad). The extent of cell proliferation was assessed by densitometry using Fiji/ImageJ software (National Institutes of Health). Note, hTERT-RPE1 cells are highly mobile and do not form distinct colonies.

### Cell-cycle tracking using hTERT-RPE1 p53[WT] and p53[KO] FUCCI cell lines

For single cell tracking, the cells were plated in 35 mm dishes with a 14 mm, 1.5 thickness cover glass window on the bottom (MatTek Corp no. P35G-1.5-14-C). For imaging, the dishes were placed in a 37 °C and 5% $CO_2$ environment chamber (Tokai Hit) on the microscope stage. Imaging was performed on an Ultraview Vox spinning disk confocal system running Volocity software (PerkinElmer) using the 20×/0.75 NA UPlanSApo objective on an Olympus IX81 inverted microscope equipped with an electron multiplying charge coupled device (EM-CCD) camera (Hamamatsu Photonics no. C9100-13). FUCCI probes and p21–GFP were imaged with 488 nm and 561 nm lasers using 100–200 ms exposures at 3–6% laser power. Brightfield reference images were also taken to visualize cell shape with 30 ms exposures. Image stacks, 24 planes at 0.9 μm spacing were collected at the time intervals indicated in the figures, and then the maximum intensity was projected and cropped in Fiji/ImageJ (National Institutes of Health) for further analysis. Before the imaging, the cells were treated with either 0.1% (v/v) DMSO as a control, 25 ng ml$^{-1}$ (82.5 nM) nocodazole, 30 nM CENP-E inhibitor, 100 nM MD-224, 2.5 μM Nutlin-3a or combinations of these chemicals for 30 min or 4 h at 37 °C, 5% $CO_2$ and then washed five times in prewarmed 37 °C full medium. After the washout, the dish was immediately placed on the microscope, and then, the mitotic cells expressing mAG-hGem, in prometaphase or metaphase, were imaged and tracked to measure G1 duration.

### Serum starvation to induce ciliogenesis

hTERT-RPE1 cells were seeded on sterile glass coverslips inside six-well culture plates. After 24 h, the cells were washed three times in serum-free medium and then incubated in serum-free medium for 40 h to allow cilia to form. The cells were then fixed in 3% (w/v) paraformaldehyde (PFA) (Sigma-Aldrich no. 158127) in PBS pH 7.4 and processed for fluorescence microscopy with staining for the ciliary marker Arl13b.

### Immunofluorescence microscopy

The cells plated on glass coverslips were washed in PBS and fixed with 3% (w/v) PFA in PBS pH 7.4 t room temperature for 15 min. Unreacted PFA was quenched by a 10 min incubation in 50 mM $NH_4Cl$ (Sigma-Aldrich no. 213330) in PBS. The cells were then washed three times in PBS, followed by permeabilization with 0.2% (v/v) Triton X-100 (Sigma-Aldrich no. X100) in PBS at room temperature for 5–10 min. The permeabilized cells were washed three times in PBS before incubating with the appropriate primary antibodies (Supplementary Methods Table 1). The antibody dilutions were performed in PBS, and the coverslips were placed cell-side down on droplets of antibody solutions, inside a humidified chamber for 1 h at room temperature. Following incubation, the cells were washed three times in PBS and incubated with the secondary AlexaFluor-conjugated antibodies (Supplementary Methods Table 1) for 1 h at room temperature as described for the primary antibody incubation. 4,6-Diamidino-2-phenylindole (Sigma-Aldrich no. D9542) was added at the same time as the secondary antibodies, to stain DNA. Following incubation, the cells were washed three times in PBS and once in water before mounting on glass microscopy slides in Mowiol 4-88 (Sigma-Aldrich no. 81381). The samples were imaged on an upright microscope (BX61, Olympus) with filter sets for DAPI, GFP/Alexa Fluor 488, Cy3/Alexa Fluor 555 and Cy5/Alexa Fluor 647 (Chroma

Technology), a 2,048 × 2,048-pixel complementary metal oxide semiconductor camera (Prime, Photometrics) and MetaMorph 7.5 imaging software (Molecular Devices). Illumination was provided by an LED light source (pE300, CoolLED Illumination Systems). The image stacks with a spacing of 0.4 μm through the cell volume were maximum intensity projected and cropped in ImageJ (National Institutes of Health).

### Data quantification

Raw western blots and imaging data were quantified using Fiji (ImageJ) (v.2.14.0). MDM2 protein half-life and the half-maximal effective concentration ($EC_{50}$) of MD-224 were calculated with GraphPad Prism v.10.3.0 using the non-linear one phase decay and $EC_{50}$ shift functions, respectively.

### Statistics and reproducibility

Statistical analysis was performed using GraphPad Prism v.10.3.0. The statistical significance was analysed using an unpaired two-tailed $t$-test, a one- or two-way analysis of variance (ANOVA), a Brown–Forsythe ANOVA, a Kruskal–Wallis test or Dunn's multiple comparison test. A Dunnett's or Tukey's multiple comparisons tests was performed following the ANOVA analyses. The graphs display the mean ± standard deviation (s.d.) or standard error of the mean (s.e.m.). Where statistical tests were performed, specific $P$ values are shown on graphs with asterisks as follows: $P ≥ 0.05$ for not significant (n.s.) and *$P < 0.05$, **$P < 0.01$, ***$P < 0.001$, ****$P < 0.0001$. No statistical methods were used to predetermine the experimental sample sizes. The sample size was chosen to balance replication and efficiency in the experiments. The exact sample size of the associated experiments is described in the figure legends. The exact number of replicates/experiments are indicated in the corresponding figures or their legends. All experiments were performed with molecular biological techniques. Randomization was not performed, since experiments were performed with independent replicates and populations of manipulated cell lines for cell culture-based experiments.

### Reporting summary

Further information on research design is available in the Nature Portfolio Reporting Summary linked to this article.

## Data availability

Source data are provided with this paper. All other data supporting the findings of this study are available from the corresponding author on reasonable request.

## Acknowledgements

We thank U. Grüneberg, L. Jansen, B. Novak, M. Srinivasan and our colleagues in the Barr group for discussion and advice, M. Attwood for help with generation and validation of the p53[KO] hTERT-RPE1 cells and P. Mikulski for assistance with RT–qPCR. Cancer Research UK programme grant award DRCRPG-May23/100006 (FAB) and Cancer Research UK Career Development Fellowship C62538/A24670 (IGS) funded this work.

## Author contributions

Conceptualization: F.A.B. Methodology: L.J.F., T.S., I.G.S. and F.A.B. Investigation: L.J.F., T.S. and C.B. Funding acquisition: F.A.B. and I.G.S. Supervision: F.A.B. Writing—original draft: F.A.B. Writing—review and editing: F.A.B., L.J.F. and T.S.

## Competing interests

The authors declare no competing interests.

## Additional information

**Extended data** is available for this paper at https://doi.org/10.1038/s41556-024-01592-8.

**Correspondence and requests for materials** should be addressed to Francis A. Barr.

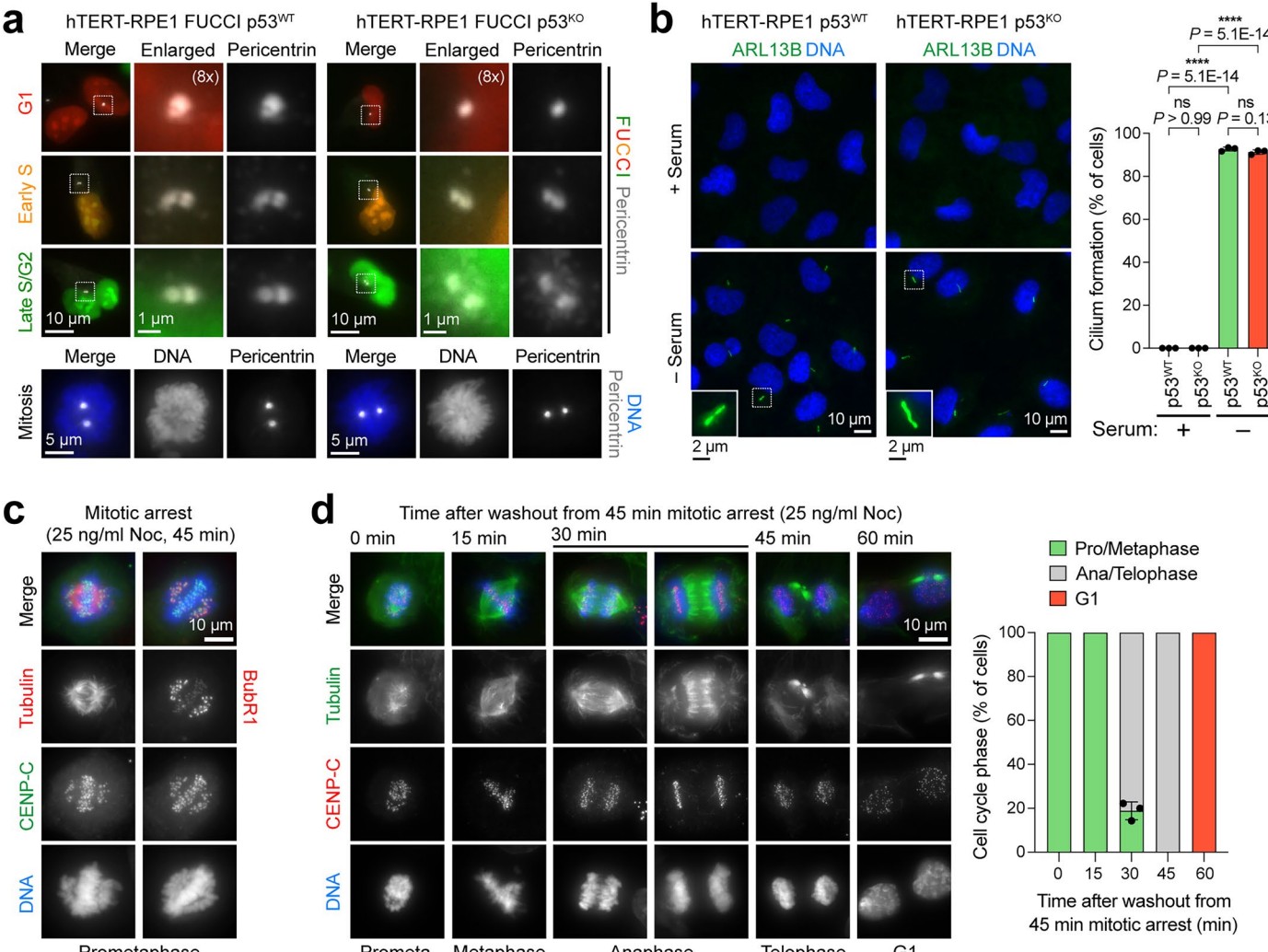

**Extended Data Fig. 1 | Low dose nocodazole activates the spindle assembly checkpoint and allows rapid release into anaphase. a**, Asynchronous hTERT-RPE1 FUCCI cells were fixed and stained for centrosomes (Pericentrin). Representative images for each cell cycle phase are shown (n = 3 independent experiments). **b**, Asynchronous hTERT-RPE1 p53$^{WT}$ or p53$^{KO}$ cells were serum starved for 40 h before fixation and staining for primary cilia (Arl13b) and DNA. The percentage of cells with cilia for both p53$^{WT}$ and p53$^{KO}$ cells are quantified in the accompanying bar graph (mean ± s.e.m.; n = 3 independent experiments; p > 0.99 p53$^{WT}$ +serum vs. p53$^{KO}$ +serum, p = 0.13 p53$^{WT}$ −serum vs. p53$^{KO}$ −serum, p = 5.1E$^{−14}$ p53$^{WT}$ +serum vs. p53$^{WT}$ −serum, p = 1E$^{−14}$ p53$^{KO}$ +serum vs. p53$^{KO}$ − serum). **c**, hTERT-RPE1 cells treated with 25 ng/ml nocodazole (Noc) for 3 h were fixed and stained for DNA, tubulin, the centromere protein CENP-C, and either tubulin or the spindle assembly checkpoint marker BubR1 (n = 3 independent experiments with similar results). **d**, hTERT-RPE1 cells treated with 25 ng/ml nocodazole (Noc) for 45 min. Nocodazole was then washed out with fresh growth medium, and the cells were imaged at the indicated timepoints as they exited mitosis and entered G1. The percentage of cells in each mitotic phase for the indicated timepoints was quantified (mean ± s.d.; n = 3 independent experiments). Data were analysed using one-way ANOVA test with Tukey's multiple comparisons (**b**) (ns, not significant; ****, p < 0.0001). Source numerical data are available in source data.

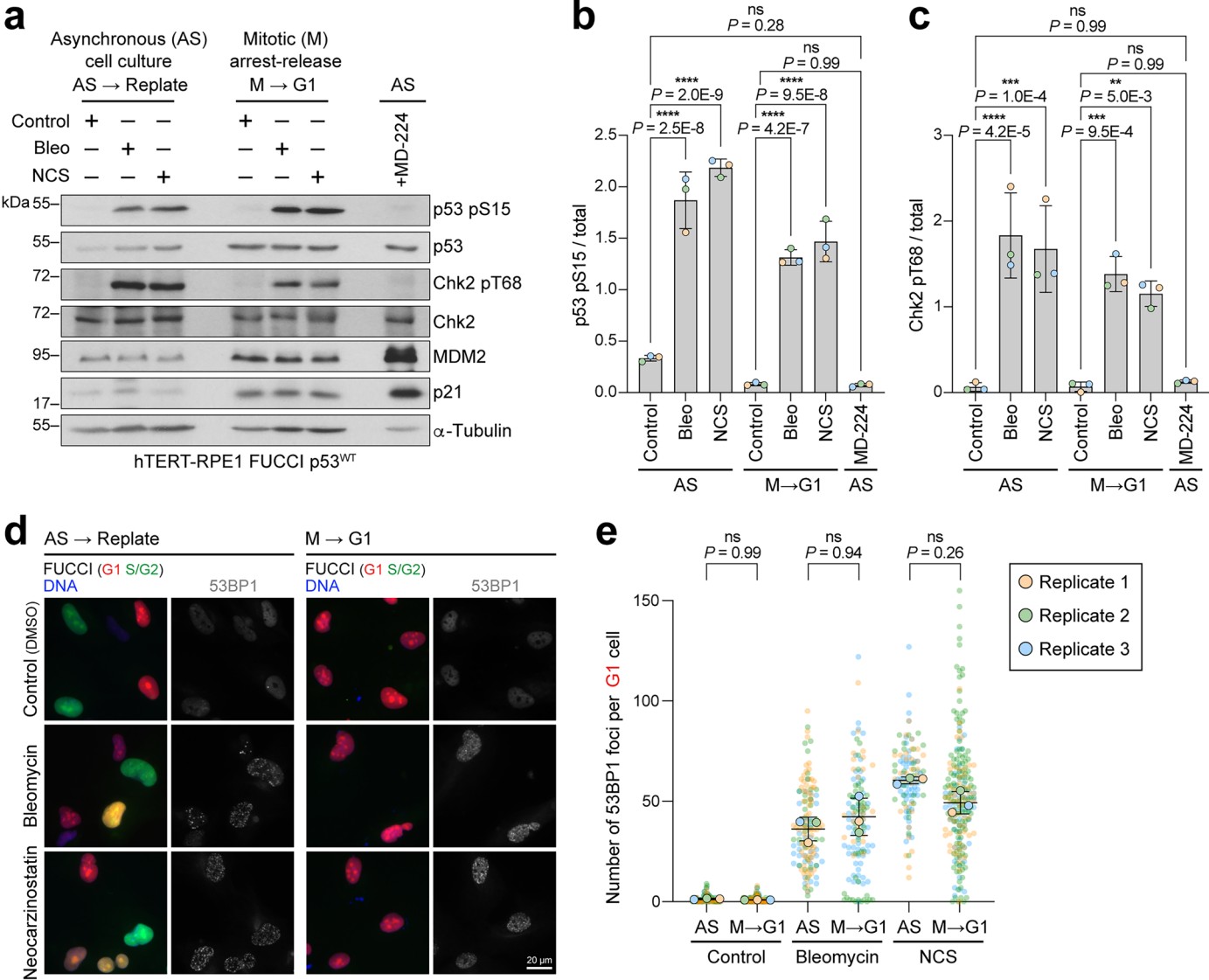

**Extended Data Fig. 2 | Delays in mitosis do not induce DNA damage in subsequent G1 cells. a**, Asynchronous (AS) hTERT-RPE1 p53^WT cell cultures were plated to mimic the conditions in Fig. 1a–c, then 17 h later the replated cells were treated with DMSO (Control), 30 µg/ml bleomycin (Bleo) or 200 ng/ml neocarzinostatin (NCS) for 1 h to induce DNA damage. To test the effect of mitotic arrest on DNA damage, hTERT-RPE1 p53^WT cells were treated with 25 ng/ml nocodazole for 18 h. Mitotic cells harvested by shake-off and nocodazole washed-out, replated, and then 17 h later treated with DMSO, Bleo or NCS for 1 h. In both cases, DNA damage induction was assessed by Western blotting (n = 3 independent experiments with similar results). **b,c**, Graphs calculated from **a** showing the quantification of phosphorylated p53 (p53 pS15) (**b**) and Chk2 (Chk2 pT68) (**c**) relative to total proteins (mean ± s.d.; n = 3 independent experiments; p53 pS15/total, p = 2.4E^−8 AS Control vs. AS Bleo, p = 2.0E^−9 AS Control vs. AS NCS, p = 4.2E^−7 M-G1 Control vs. M-G1 Bleo, p = 9.5E^−8 M-G1 Control vs. M-G1 NCS,

p = 0.99 M-G1 Control vs. AS MD-224, p = 0.28 AS Control vs. AS MD-224; Chk2 pT68/total, p = 4.2E^−5 AS Control vs. AS Bleo, p = 1.0E^−4 AS Control vs. AS NCS, p = 9.5E^−4 M-G1 Control vs. M-G1 Bleo, p = 5.0E^−3 M-G1 Control vs. M-G1 NCS, p = 0.99 M-G1 Control vs. AS MD-224, p = 0.99 AS Control vs. AS MD-224). **d**, hTERT-RPE1 p53^WT FUCCI cells treated as in **a** were stained for 53BP1. The FUCCI marker was directly visualised using fluorescence microscopy. **e**, Scatter plots of the number of 53BP1 foci in G1 cells from **d**, showing individual cells with the mean values from 3 independent experiments (mean ± s.d.; p = 0.99 AS Control vs. M-G1 Control, p = 0.93 AS Bleomycin vs. M-G1 Bleomycin, p = 0.25 AS NCS vs. M-G1 NCS). Data were analysed using one-way ANOVA test with Tukey's multiple comparisons (**b,c**) and two-way ANOVA test with Šídák's multiple comparisons test (**e**) (ns, not significant; **, p < 0.01; ***, p < 0.001; ****, p < 0.0001). Source numerical data and unprocessed blots are available in source data.

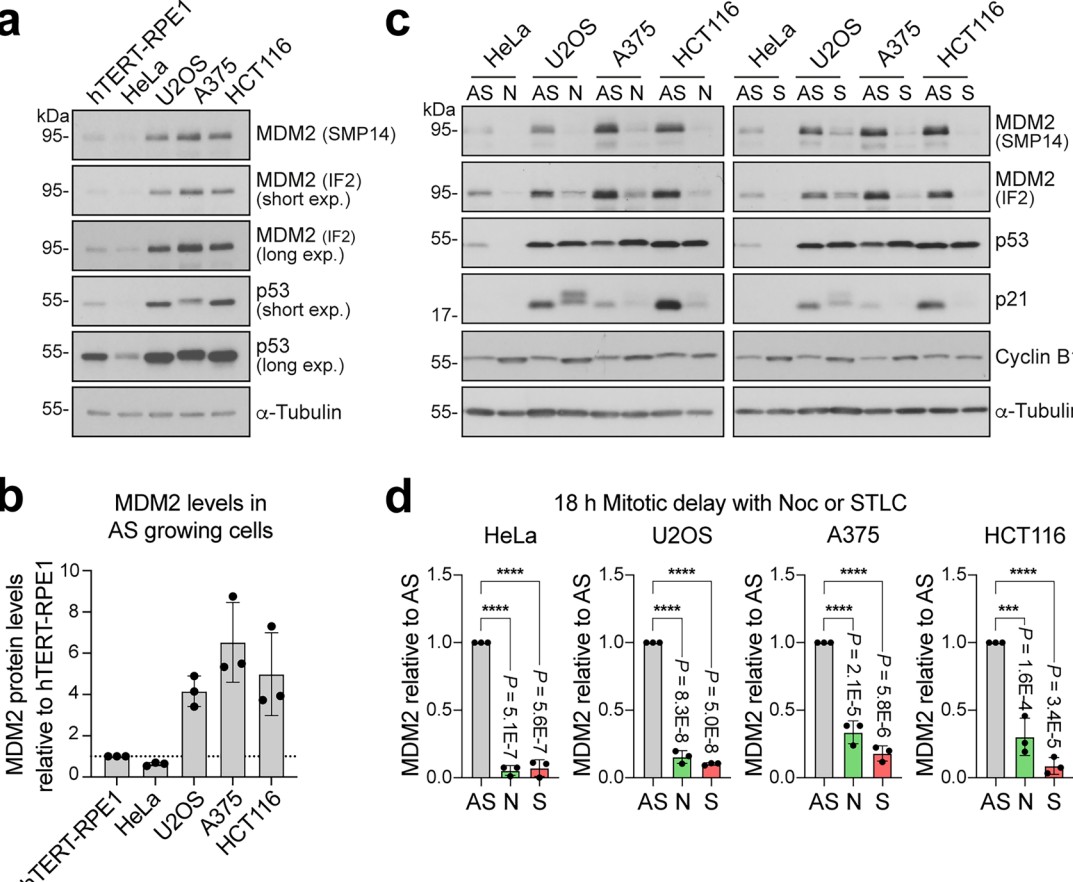

**Extended Data Fig. 3 | Reduction in the amount of MDM2 during mitosis in different human cell lines. a**, The amounts of MDM2, p53 and p21 were measured in hTERT-RPE1 p53$^{WT}$, HeLa, U2OS, A375 and HCT116 cell lines under standard culture conditions using Western blotting. α-Tubulin is a loading control. **b**, Relative amounts of MDM2 between cell lines are plotted (mean ± s.e.m.; n = 3 independent experiments). **c**, The amount of MDM2 was measured by Western blotting in asynchronous (AS), nocodazole (N) or S-trityl L-cysteine (STLC, S)-arrested mitotic HeLa, U2OS, A375 and HCT116 cell lines.

**d**, Quantification of MDM2 in HeLa, U2OS, A375 and HCT116 cell lines using data from **c**. Values are expressed relative to the AS control for that cell line (mean ± s.e.m.; n = 3 independent experiments; HeLa p = 5.1E$^{-7}$ AS vs. N, p = 5.6E$^{-7}$ AS vs. S; U2OS p = 8.3E$^{-8}$ AS vs. N, p = 5.0E$^{-8}$ AS vs. S; A375, p = 2.1E$^{-5}$ AS vs. N, p = 5.8E$^{-6}$ AS vs. S; HCT116 cells, p = 1.6E$^{-4}$ AS vs. N, p = 3.4E$^{-5}$ AS vs. S). Data were analysed using one-way ANOVA test with Tukey's multiple comparisons (**d**) (***, p < 0.001; ****, p < 0.0001). Source numerical data and unprocessed blots are available in source data.

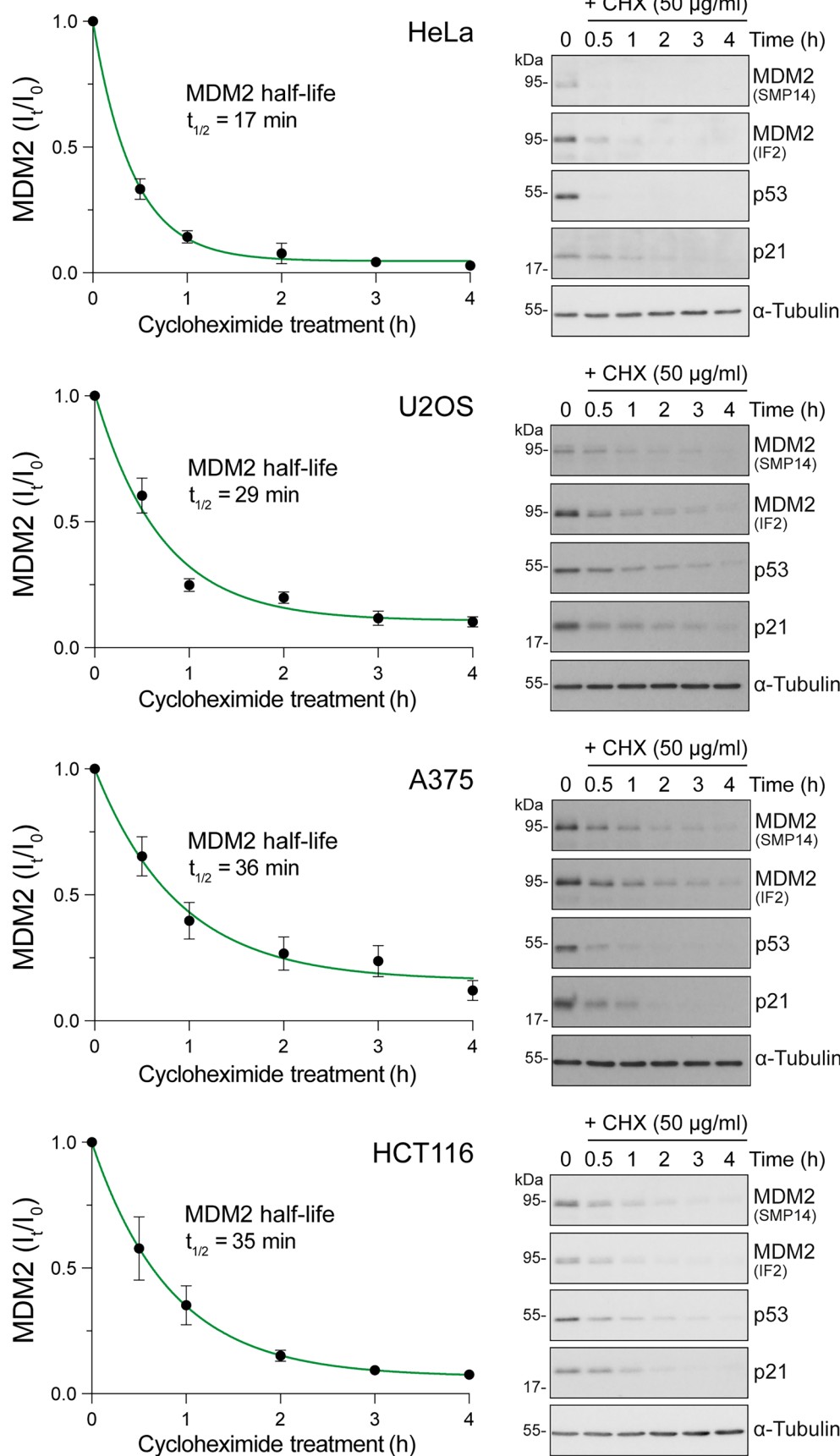

**Extended Data Fig. 4 | Short half-life is a general property of MDM2 in multiple human cell lines.** MDM2 half-life was measured in HeLa, U2OS, A375 and HCT116 cell lines by adding 50 µg/ml cycloheximide (CHX) to block protein synthesis, and then collecting samples up to 4 h. MDM2 half-life is indicated in the graph (mean ± s.d.; n = 3 independent experiments), with representative Western blots. MDM2 half-life was calculated from the curve fit. Source numerical data and unprocessed blots are available in source data.

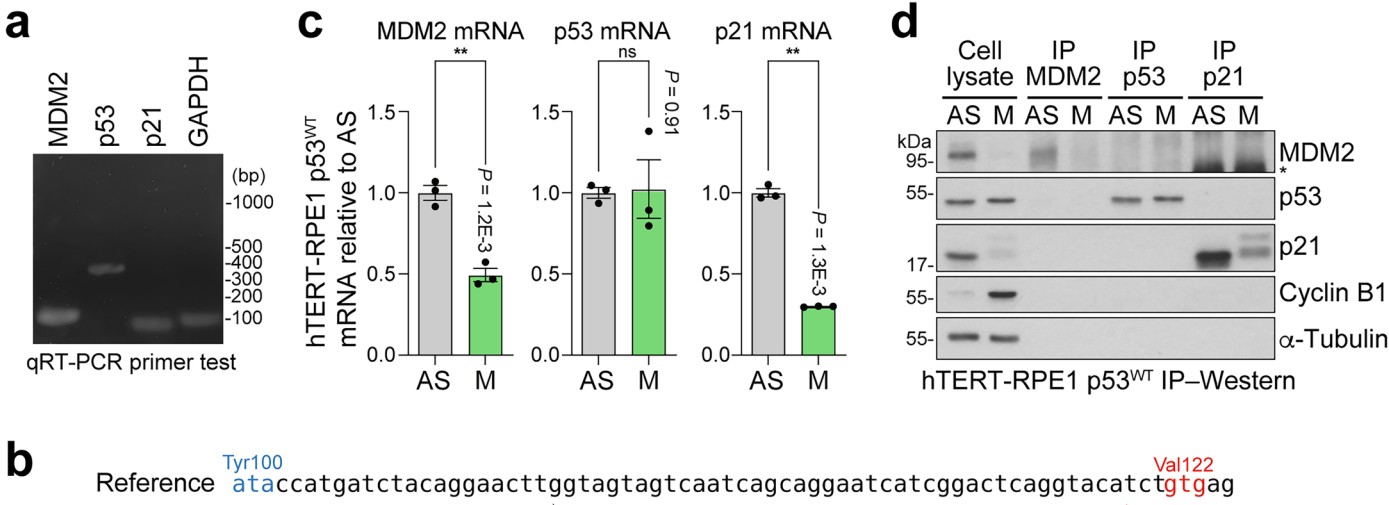

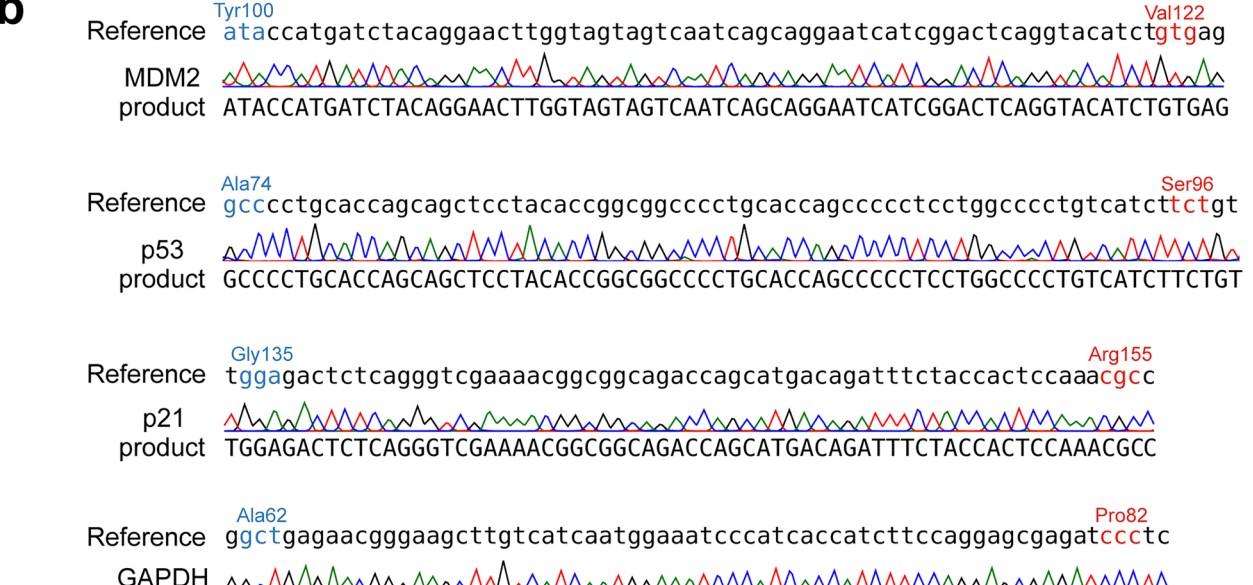

**Extended Data Fig. 5 | RT-qPCR reveals reduction of MDM2 mRNA in mitosis.**
**a**, Agarose gel showing RT-qPCR products obtained using diagnostic primers for MDM2, p53, p21 and GAPDH. 2 independent experiments showed similar results. **b**, Reference and confirmed DNA sequences for the MDM2, p53, p21 and GAPDH RT-qPCR products. **c**, RT-qPCR from asynchronous hTERT-RPE1 p53^WT cells (AS) and cells arrested for 18 h in mitosis with 25 ng/ml nocodazole (M). All mRNA levels are expressed relative to the AS control samples and normalised against GAPDH (mean ± s.e.m.; n = 3 biological repeats each with 3 technical replicates; AS vs. M for MDM2 mRNA, p = 1.1E$^{-3}$, p53 mRNA, p = 0.91, p21 mRNA, p = 1.1E$^{-5}$). **d**, Immunoprecipitation of MDM2, p53 and p21 from AS and M conditions used in **c**. 3 independent experiments showed similar results. The asterisk (*) denotes a non-specific band. Data were analysed using unpaired two-tailed t-test with Welch's correction (**c**) (ns, not significant; **, p < 0.01). Source numerical data and unprocessed blots are available in source data.

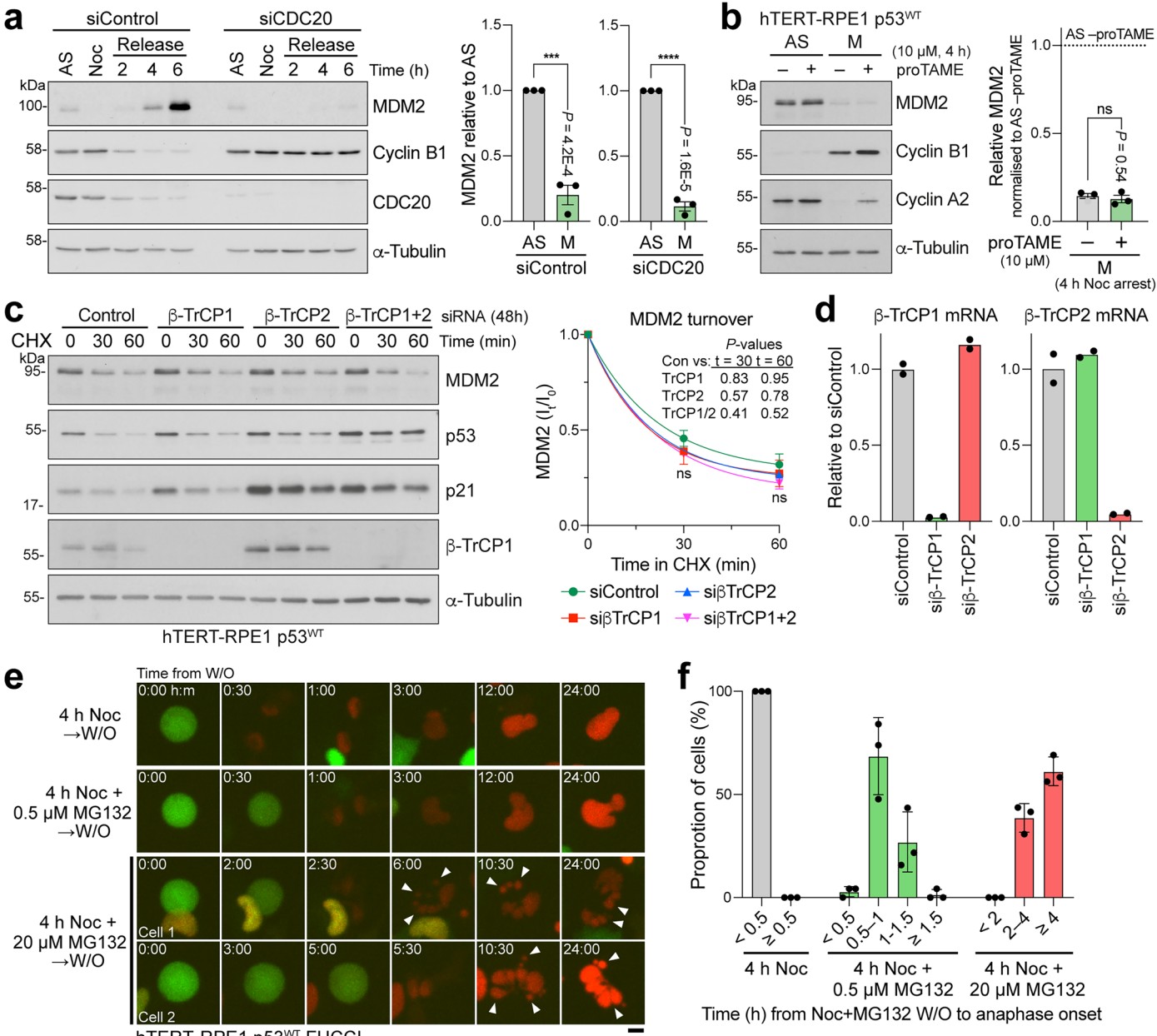

**Extended Data Fig. 6 | APC/C$^{CDC20}$ and SCF$^{\beta\text{-TRCP}}$ do not regulate MDM2 stability in mitosis. a**, HeLa cells depleted of the APC/C co-activator CDC20 (siCDC20) or treated with a non-targeting control (siControl) were arrested in mitosis for 18 h with 100 ng/ml nocodazole (Noc). Nocodazole was washed out to test for CDC20 depletion, confirmed by cyclin B1 stabilisation for up to 6 h in the siCDC20 treated cells. Untreated asynchronous cells (AS) were taken as a control. Compared to asynchronous (AS) cells, MDM2 was lost during the mitotic arrest independently of CDC20, and did not recover in the washout in CDC20-depleted cells due to the sustained mitotic arrest. MDM2 recovered in the control cells due to resumption of protein synthesis during mitotic exit and G1 entry (quantified on the right; mean ± s.e.m.; n = 3 independent experiments; p = 4.2E$^{-4}$ siControl AS vs. siControl M, p = 1.6E$^{-5}$ siCDC20 AS vs. siCDC20 M). **b**, MDM2 levels were measured in AS and 25 ng/ml nocodazole-arrested mitotic cells by Western blot after 4 h treatment with 10 μM of the APC/C inhibitor proTAME (+). Solvent treated (−) cells were used as a control. Relative mitotic MDM2 levels for each condition are plotted on the right (mean ± s.e.m.; n = 3 independent experiments; p = 0.54). **c**, MDM2 stability was measured by Western blot following 30- or 60-minutes treatment with 50 μg/ml cycloheximide (CHX) in cells depleted of the SCF subunits β-TrCP1, β-TrCP2, or β-TrCP1 and 2

together. Non-targeting siRNA was used as a control. Relative MDM2 half-lives for each condition are plotted on the right (mean ± s.e.m.; n = 3 independent experiments; p = 0.83 siControl 30 min vs. siβTrCP1 30 min, p = 0.57 siControl 30 min vs. siβTrCP2 30 min, p = 0.41 siControl 30 min vs. siβTrCP1 + 2 30 min, p = 0.95 siControl 60 min vs. siβTrCP1 60 min, p = 0.78 siControl 60 min vs. siβTrCP2 60 min, p = 0.52 siControl 60 min vs. siβTrCP1 + 2 60 min). **d**, RT-qPCR was used to measure the levels of β-TRCP1 and β-TRCP2 mRNAs in cells depleted of β-TrCP1 or β-TrCP2 by siRNA to confirm successful knockdown. Signals were normalised to GAPDH mRNA and expressed relative to the Control sample, and the mean is plotted in the bar graphs (n = 2 biological repeats each with 3 technical replicates). **e**, Representative images of hTERT-RPE1 p53$^{WT}$ FUCCI cells arrested in mitosis for 4 h with 25 ng/ml nocodazole with either DMSO control or the indicated doses of MG132. Nocodazole and MG132 were washed out and cells tracked to monitor their miotic exit kinetics. Arrowheads indicate nuclear morphology defects. **f**, Quantification of mitotic exit kinetics from **e** is plotted (mean ± s.d.; n = 3 independent experiments). Data were analysed using unpaired two-tailed t-test (**a**,**b**) or two-way ANOVA test with Tukey's multiple comparisons (**c**) (ns, not significant; ***, p < 0.001; ****, p < 0.0001). Source numerical data and unprocessed blots are available in source data.

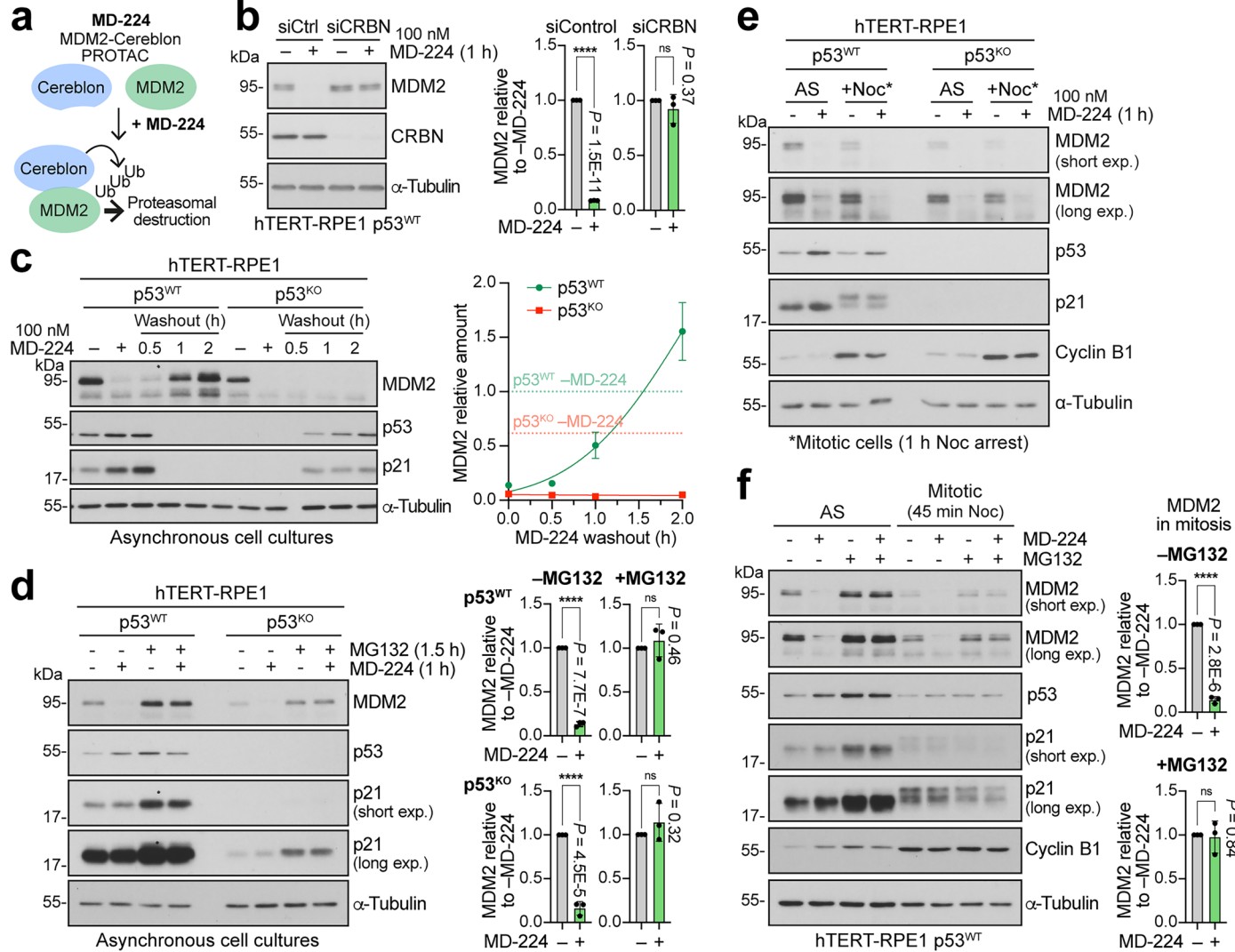

**Extended Data Fig. 7 | MD-224 results in rapid and reversible MDM2 destruction in hTERT-RPE1 cells. a**, A cartoon showing the mode of action for MD-224. **b**, hTERT-RPE1 p53$^{WT}$ cells depleted of CRBN with siRNA were treated with solvent (−) or 100 nM MD-224 (+) for 1 h, and Western blotted. MD-224 requires CRBN for MDM2 destruction. Non-targeting siRNA was used as a control. Relative depletion of MDM2 upon MD-224 treatment is plotted for each siRNA condition on the right (mean ± s.d.; n = 3 independent experiments; p = 1.5E$^{-11}$ siControl −MD-224 vs. siControl +MD-224, p = 0.37 siCRBN −MD-224 vs. siCRBN +MD-224). **c**, hTERT-RPE1 p53$^{WT}$ or p53$^{KO}$ cells were treated with 100 nM MD-224 for 1 h (+) or a solvent control (−). MD-224 was then washed out with fresh growth medium for 0.5 to 2 h and Western blotted for MDM2, p53 and p21. Tubulin was used as a loading control. Relative recovery of MDM2 following MD-224 washout is plotted on the right for each cell line (mean ± s.e.m.; n = 3 independent experiments). **d**, The amount of MDM2 was measured by Western blotting in asynchronous (AS) hTERT-RPE1 p53$^{WT}$ and p53$^{KO}$ cells treated for 1.5 h with (+) or without (−) the proteasome inhibitor MG132 (20 μM) and for 1 h with (+) or without (−) 100 nM MD-224. Relative depletion of MDM2 in each condition is plotted on the right for each cell line (mean ± s.d.; n = 3 independent

experiments; p53$^{WT}$ cells, p = 7.7E$^{-7}$ −MG132 −MD-224 vs. −MG132 + MD-224, p = 0.46 + MG132 −MD-224 vs. +MG132 + MD-224; p53$^{KO}$ cells, p = 4.5E$^{-5}$ −MG132 −MD-224 vs. −MG132 + MD-224, p = 0.32 + MG132 −MD-224 vs. +MG132 + MD-224). **e**, The amount of MDM2 was measured in asynchronous cells (AS) or mitotic hTERT-RPE1 p53$^{WT}$ and p53$^{KO}$ cells arrested for 1 h with 25 ng/ml nocodazole ( + Noc) in the presence (+) or absence (−) of 100 nM MD-224. MDM2 was detected by Western blot using two different antibodies. n = 5 independent experiments with similar results (see Fig. 4). **f**, The amount of MDM2 was measured using Western blotting in asynchronous cells (AS) or mitotic hTERT-RPE1 p53$^{WT}$ cells arrested for 45 min with 25 ng/ml nocodazole ( + Noc) in the presence (+) or absence (−) of 100 nM MD-224 and the proteasome inhibitor MG132 (20 μM). Relative MDM2 levels for each mitotic condition are plotted on the right (mean ± s.d.; n = 3 independent experiments; p = 2.8E$^{-6}$ −MG132 −MD-224 vs. −MG132 + MD-224, p = 0.84 + MG132 −MD-224 vs. +MG132 + MD-224). Data were analysed using unpaired two-tailed t-test (**b,d,f**) (ns, not significant; ****, p < 0.0001). Source numerical data and unprocessed blots are available in source data.

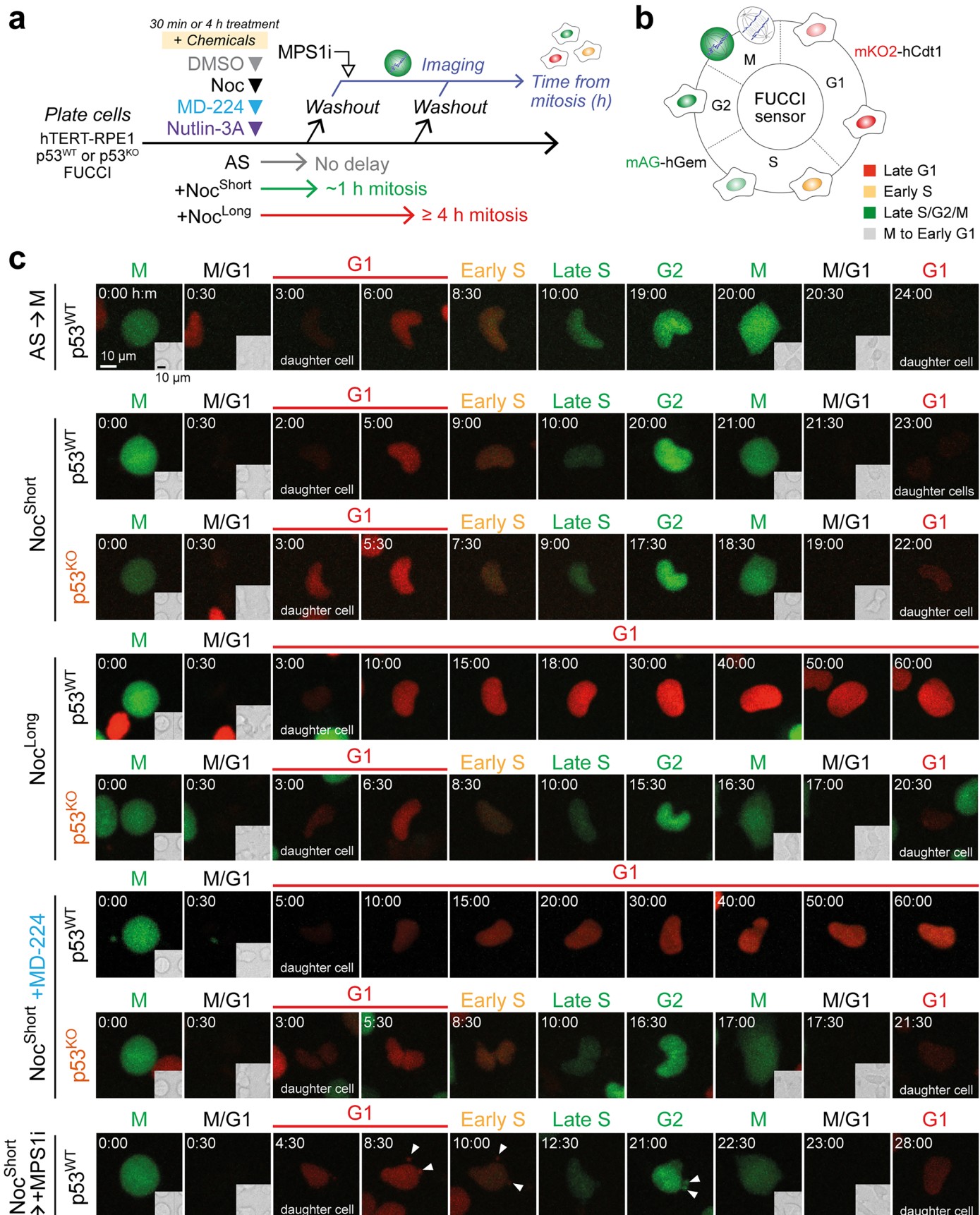

**Extended Data Fig. 8 | hTERT-RPE1 FUCCI cell lines progress through G1 into S-phase when mitosis is not delayed. a**, Experimental design used for Fig. 5. **b**, The properties of the FUCCI sensor. **c**, Representative images of hTERT-RPE1 p53[WT] and p53[KO] FUCCI cells for the conditions described in Fig. 5a–c; n = 3 independent experiments per condition. Arrowheads in Noc[Short] → MPS1i indicate micronuclei.

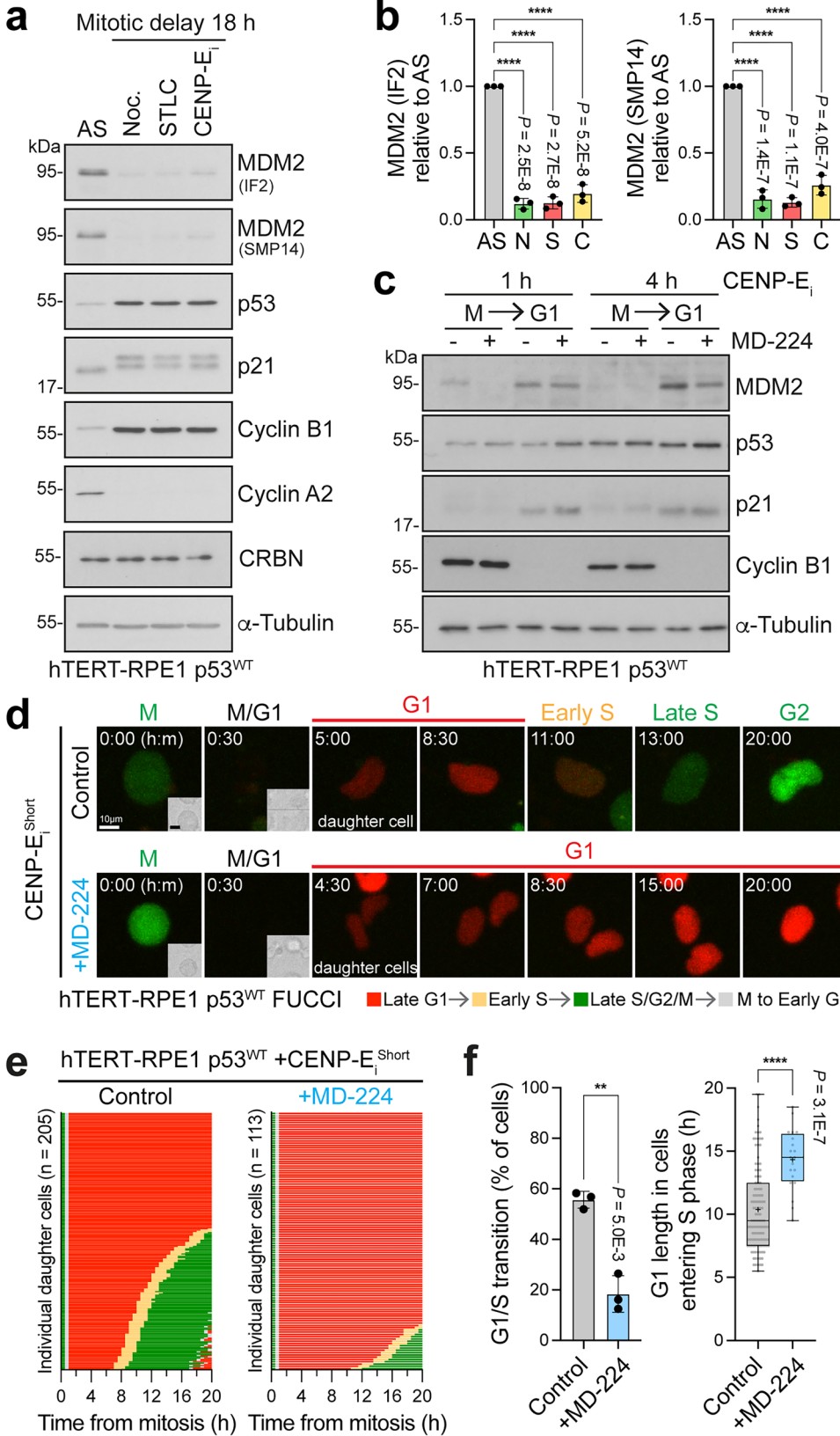

**Extended Data Fig. 9 | See next page for caption.**

**Extended Data Fig. 9 | Mitotic delays induced by CENP-E inhibition trigger G1 arrest. a**,**b**, The amount of MDM2 was measured by Western blotting in asynchronous (AS) or mitotic hTERT-RPE1 p53$^{WT}$ cells, arrested in mitosis for 18 h with 100 ng/ml nocodazole (N), 5 μM S-trityl L-cysteine (S), or 300 nM CENP-E inhibitor (C). Relative MDM2 levels across the different conditions were quantified for two distinct MDM2 antibodies (**b**) (mean ± s.d.; n = 3 independent experiments; MDM2 IF2, p = 2.5E$^{-8}$ AS vs. N, p = 2.7E$^{-8}$ AS vs. S), p = 5.2E$^{-8}$ AS vs. C; MDM2 SMP14, p = 1.4E$^{-7}$ AS vs. N, p = 1.1E$^{-7}$ AS vs. S, p = 4.0E$^{-7}$ AS vs. C). **c**, Western blot of cyclin B1 positive mitotic (M) and cyclin B1 negative (G1) hTERT-RPE1 p53$^{WT}$ cells arrested for either 1 h or 4 h in mitosis with 30 nM CENPE-inhibitor (CENPE$_i$), in the absence (−) or presence (+) of 100 nM MD-224 for the final hour. 3 independent experiments showed similar results. **d**, Representative images of hTERT-RPE1 p53$^{WT}$ passing from mitosis into the following cell cycle after arrest in mitosis for 45 min with 30 nM CENP-E inhibitor (CENP-E$_i$$^{Short}$) in the absence

or presence of 100 nM MD-224 prior to washout. The mitotic cells were tracked after washout and then imaged. **e**,**f**, Single cell traces of hTERT-RPE1 p53$^{WT}$ FUCCI cells treated as in **d** passing from mitosis into the following cell cycle. For each condition, the number of individual daughter cells analysed are indicated. The percentage of cells undergoing the G1/S transition is plotted (mean ± s.d.), and the duration of G1 in those cells that entered the next S-phase is shown in a box and whiskers plot (median, 25th and 75th percentiles, and whiskers extending to minimum and maximum values) with the mean (+) for the different conditions; n = 3 independent experiments per condition; p = 5.0E$^{-3}$ G1/S transition Control vs. +MD-224, p = 3.1E$^{-7}$ G1 length in cells entering S phase Control vs. +MD-224) (**d**–**f**). Data were analysed using one-way ANOVA test with Tukey's multiple comparisons (**b**) or unpaired two-tailed t-test with Welch's correction (**f**) (ns, not significant; **, p < 0.01; ****, p < 0.0001). Source numerical data and unprocessed blots are available in source data.

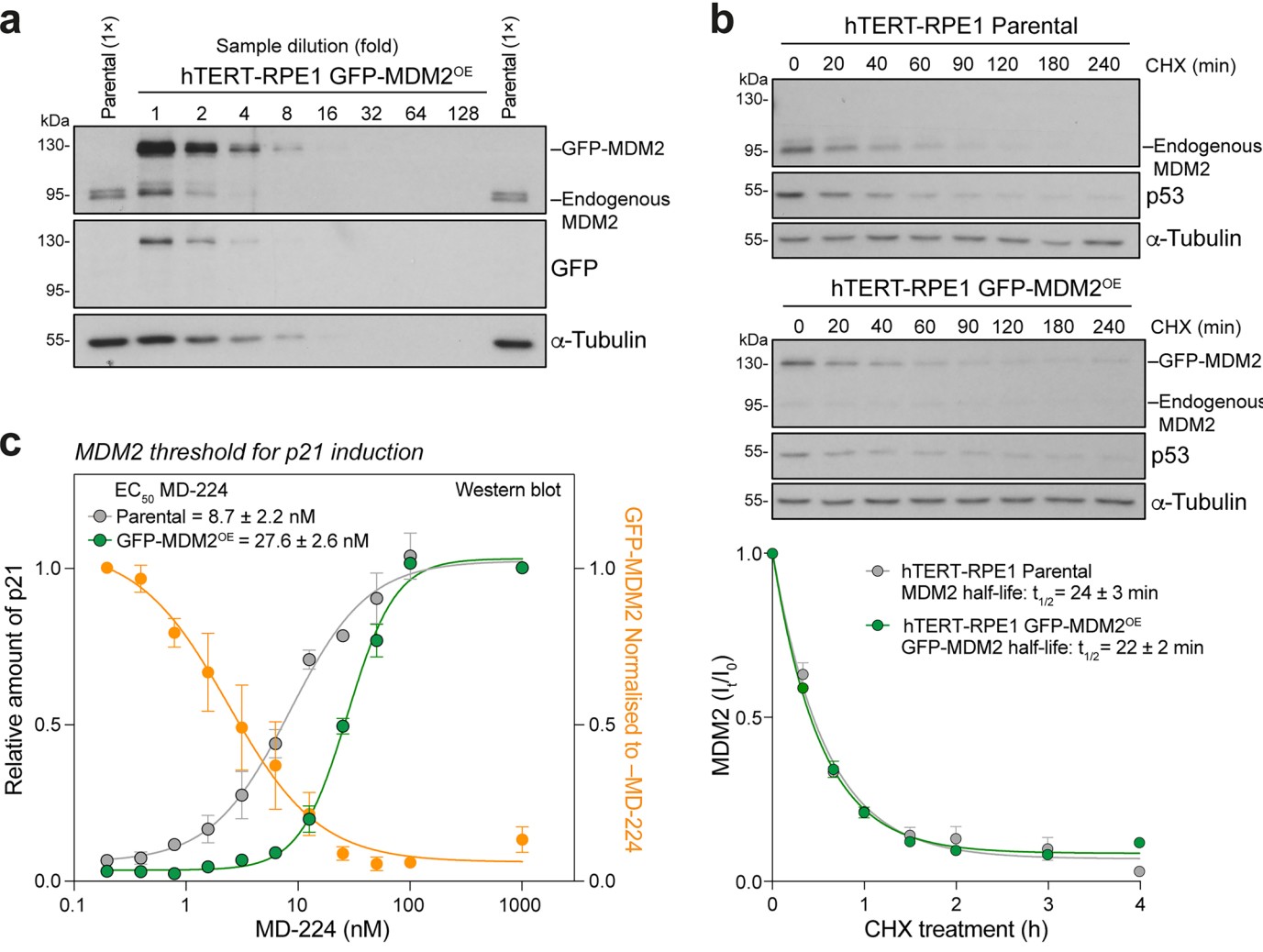

**Extended Data Fig. 10 | Additional characterisation of hTERT1-RPE1 GFP-MDM2 overexpressing cells. a**, Western blot of undiluted parental hTERT-RPE1 p53[WT] and a dilution series of GFP-MDM2[OE] cell lysates. GFP-MDM2 level relative to endogenous MDM2 shown in Fig. 6a was measured from this data; n = 3 independent experiments with similar results. **b**, MDM2 and GFP-MDM2 half-lives were measured using cycloheximide treatment of parental hTERT-RPE1 p53[WT] and GFP-MDM2[OE] cells for up to 4 h. Samples were Western blotted and MDM2 levels measured and plotted in the graphs. Half-lives were estimated using a 1-phase curve fit; n = 3 independent experiment. **c**, Titration of MD-224 to reveal the threshold MDM2 concentration (orange line, circles) required for p21 induction. Total p21 levels were measured by Western blot in hTERT-RPE1 p53[WT] (grey line, circles) and GFP-MDM2[OE] cells (green line, circles) (mean ± s.e.m.; n = 3 independent experiments with similar results). GFP-MDM2 levels in GFP-MDM2[OE] cells were also measured (orange line, circles) (mean ± s.e.m.; n = 3 independent experiments with similar results). Source numerical data and unprocessed blots are available in source data.

# Reporting Summary

## Statistics

For all statistical analyses, confirm that the following items are present in the figure legend, table legend, main text, or Methods section.

| n/a | Confirmed | |
|---|---|---|
| ☐ | ☒ | The exact sample size (*n*) for each experimental group/condition, given as a discrete number and unit of measurement |
| ☐ | ☒ | A statement on whether measurements were taken from distinct samples or whether the same sample was measured repeatedly |
| ☐ | ☒ | The statistical test(s) used AND whether they are one- or two-sided<br>*Only common tests should be described solely by name; describe more complex techniques in the Methods section.* |
| ☒ | ☐ | A description of all covariates tested |
| ☒ | ☐ | A description of any assumptions or corrections, such as tests of normality and adjustment for multiple comparisons |
| ☐ | ☒ | A full description of the statistical parameters including central tendency (e.g. means) or other basic estimates (e.g. regression coefficient) AND variation (e.g. standard deviation) or associated estimates of uncertainty (e.g. confidence intervals) |
| ☐ | ☒ | For null hypothesis testing, the test statistic (e.g. *F*, *t*, *r*) with confidence intervals, effect sizes, degrees of freedom and *P* value noted<br>*Give P values as exact values whenever suitable.* |
| ☒ | ☐ | For Bayesian analysis, information on the choice of priors and Markov chain Monte Carlo settings |
| ☒ | ☐ | For hierarchical and complex designs, identification of the appropriate level for tests and full reporting of outcomes |
| ☒ | ☐ | Estimates of effect sizes (e.g. Cohen's *d*, Pearson's *r*), indicating how they were calculated |

*Our web collection on statistics for biologists contains articles on many of the points above.*

## Software and code

Policy information about availability of computer code

| Data collection | Plate imaging: Bio-Rad Gel Doc XR+ imaging system (Bio-Rad, 1708195EDU);<br>Immunofluorescence: MetaMorph 7.5 imaging software (Molecular Devices);<br>Live cell imaging: Volocity 6 software (PerkinElmer) |
|---|---|
| Data analysis | Statistical analysis: GraphPad Prism v.10.3.0 (GraphPad Software);<br>Image analysis: Fiji/Image J (v.2.14.0) software (National Institutes of Health); Image Lab 6 (Bio-Rad) |

For manuscripts utilizing custom algorithms or software that are central to the research but not yet described in published literature, software must be made available to editors and reviewers. We strongly encourage code deposition in a community repository (e.g. GitHub). See the Nature Portfolio guidelines for submitting code & software for further information.

## Data

Policy information about availability of data

All manuscripts must include a data availability statement. This statement should provide the following information, where applicable:

- Accession codes, unique identifiers, or web links for publicly available datasets
- A description of any restrictions on data availability
- For clinical datasets or third party data, please ensure that the statement adheres to our policy

All source data (numerical data and uncropped blots) relating to the manuscript are provided with this manuscript.

# Research involving human participants, their data, or biological material

Policy information about studies with human participants or human data. See also policy information about sex, gender (identity/presentation), and sexual orientation and race, ethnicity and racism.

| | |
|---|---|
| Reporting on sex and gender | N/A |
| Reporting on race, ethnicity, or other socially relevant groupings | N/A |
| Population characteristics | N/A |
| Recruitment | N/A |
| Ethics oversight | N/A |

Note that full information on the approval of the study protocol must also be provided in the manuscript.

# Field-specific reporting

Please select the one below that is the best fit for your research. If you are not sure, read the appropriate sections before making your selection.

☒ Life sciences          ☐ Behavioural & social sciences          ☐ Ecological, evolutionary & environmental sciences

For a reference copy of the document with all sections, see nature.com/documents/nr-reporting-summary-flat.pdf

# Life sciences study design

All studies must disclose on these points even when the disclosure is negative.

| | |
|---|---|
| Sample size | No statistical methods were used to pre-determine the experimental sample sizes. Sample size was chosen based on accepted standards in the field, as outlined by Lord, S.J. et al, (2020), "SuperPlots: Communicating reproducibility and variability in cell biology", JCB, and Zweifach, A. (2024), "Determining how many cells to average for statistical testing of microscopy experiments", JCB. The exact sample size of the associated experiments are described in the Figure legends. |
| Data exclusions | No data were excluded from the analyses. |
| Replication | The exact number of independent replicates/experiments, at least 3, are indicated in the corresponding figures or their legends. All effects observed were both reproducible and penetrant. |
| Randomization | All experiments were performed with molecular biological techniques. Experiments were performed with independent replicates and populations of manipulated cell lines for cell culture-based experiments. Microscopy images in each sample were acquired from randomly selected areas. |
| Blinding | The investigators were not blinded during data collection or analysis. Image and Western blot data was analysed systematically using ImageJ/FIJI with stastical analysis in GraphPad Prism, and we determined no blinding was required. |

# Reporting for specific materials, systems and methods

We require information from authors about some types of materials, experimental systems and methods used in many studies. Here, indicate whether each material, system or method listed is relevant to your study. If you are not sure if a list item applies to your research, read the appropriate section before selecting a response.

## Materials & experimental systems

| n/a | Involved in the study |
|---|---|
| ☐ | ☒ Antibodies |
| ☐ | ☒ Eukaryotic cell lines |
| ☒ | ☐ Palaeontology and archaeology |
| ☒ | ☐ Animals and other organisms |
| ☒ | ☐ Clinical data |
| ☒ | ☐ Dual use research of concern |
| ☒ | ☐ Plants |

## Methods

| n/a | Involved in the study |
|---|---|
| ☒ | ☐ ChIP-seq |
| ☒ | ☐ Flow cytometry |
| ☒ | ☐ MRI-based neuroimaging |

# Antibodies

| | |
|---|---|
| Antibodies used | Supplementary Method Table 1 describes all antibodies used for WB, IF and IP in this study. |
| Validation | All commercial primary antibodies were validated by the supplier using human cells, in multiple independent previous studies, and in this study (please see below):<br>MDM2 (IF2) Millipore mAb OP46 Mouse - validated by siRNA - not included in the manuscript but available on request. Also validated by MD-224 PROTAC treatment (Figure 4b,5g; Extended Data Figure 7b-f).<br>MDM2 (SMP14) Abcam mAb Ab3110 Mouse - validated by siRNA - not included in the manuscript but available on request.<br>MDM2 (D1V2Z) CST mAb 86934 Rabbit - validated by siRNA - not included in the manuscript but available on request.<br>p53 (DO-7) CST mAb 48818 Mouse - validated by KO (Figure 4b).<br>p53 (7F5) CST mAb 2527 Rabbit - validated by KO - not included in the manuscript but available on request.<br>p21 (12D1) CST mAb 2947 Rabbit - validated in Figure 4b (p21 signal goes down when its transcription factor p53 is KO'd). Also confirmed by siRNA and KO - not included in the manuscript but available on request.<br>p21 (CP36/CP74) Millipore mAb 05-345 Mouse - validated by KO - not included in the manuscript but available on request.<br>UBE2D2/UbcH5b Novis Bio pAb NPB1-81769 Rabbit - validated in Figure 3e by siRNA.<br>UBE2D3/UbcH5C (D60E2) CST mAb 4330 Rabbit - validated in Figure 3e by siRNA.<br>CENP-C MBL pAb PD030 Guinea Pig - validated in Extended Data Figure 1c-d - stains centromeres.<br>BubR1 Millipore pAb MAB3612 Mouse - validated in Extended Data Figure 1c - stains checkpoint-activated kinetochores.<br>Cyclin B1 (GSN3) Millipore mAb 8A5D12 Mouse - validated in Figure 4b - present in G2/M cells, absent in G1 cells, and in Extended Data Figure 6a-b - mitotic destruction rescued by APC/C inhibition.<br>Cyclin A2 (E399) Abcam mAb Ab32498 Rabbit - validated in Extended Data Figure 6a-b - absent in mitotic cells, rescued by APC/C inhibition.<br>CRBN (D8H3S) CST mAb 71810 Rabbit - validated in Extended Data Figure 7b by siRNA.<br>α-Tubulin (DM1A) Sigma mAb 4394 Rabbit - validated in Extended Data Figure 1c-d - stains spindle microtubules.<br>Cdc20 Proteintech pAb 10252-1-AP Rabbit - validated in Extended Data Figure 6a by siRNA.<br>β-TrCP1 (D13F10) CST mAb 4394 Rabbit - validated in Extended Data Figure 6c by siRNA.<br>GFP (7.1 and 13.1) Roche mAb 11814460001 Mouse - validated in Figure 3b, 6a,c and Extended Data Figure 10a-b - band appears when GFP-tagged transgenes are expressed.<br>p53BP1 Novus Bio pAb NB100-304 Rabbit - validated by KO - not included in the manuscript but available on request. Also labels p53BP1-positive foci in response to DNA damage stimuli, as expected (Extended Data Figure 2d).<br>phospho-p53 (Ser15) CST pAb 9284 Rabbit - validated in Extended Data Figure 2a - induced upon DNA damage. Also validated by KO - not included in the manuscript but available on request.<br>phospho-Chk2 (Thr68) CST pAb 2661 Rabbit - validated in Extended Data Figure 2a - induced upon DNA damage. Also validated by ATM/ATR inhibitor treatment - not included in the manuscript but available on request.<br>Chk2 CST pAb 2662 Rabbit - extensively validated in the literature - not further validated.<br>Arl13b Proteintech pAb 17711-1-AP Rabbit - validated in Extended Data Figure 1b - stains primary cilia.<br>Pericentrin - validated in Extended Data Figure 1a - stains centrosomes (1 in G1 cells, 2 in S/G2/M cells). |

# Eukaryotic cell lines

Policy information about cell lines and Sex and Gender in Research

| | |
|---|---|
| Cell line source(s) | Original hTERT-RPE1 (#CRL-4000), HeLa (#CRL-2.2), A375 (#CRL-1619), U2OS (#HTB-96), HEK293T (#CRL-3216), HCT116 (#CCL-247) cell lines were purchased from ATCC.<br>The hTERT-RPE1 p21-GFP cell line was a gift from Alexis R. Barr.<br>Other cell lines generated in this study (e.g. p53 KO hTERT-RPE1 and FUCCI hTERT-RPE1) were derived from these sources. |
| Authentication | hTERT-RPE1 parental and p53 KO cell lines were STR profiled and authenticated by ATCC.<br>HeLa, HCT116, hTERT-RPE1 FUCCI, and hTERT-RPE1 p21-GFP cell lines were STR profiled and authenticated by NorthGene™.<br>A375, HEK293T and U2OS cell lines purchased from ATCC were not further authenticated. |
| Mycoplasma contamination | Mycoplasma negative status of cell lines was confirmed using the EZ-PCR Mycoplasma Test Kit with internal control (K1-0210, Geneflow). |
| Commonly misidentified lines<br>(See ICLAC register) | The cell lines used in our studies are not on the list as commonly misidentified lines. |

