## [Peer Review File · Nature Cell Biology]

MDM2 functions as a timer reporting the length of mitosis

Corresponding Author: Professor Francis Barr

This manuscript has been previously reviewed at another journal. This document only contains information relating to versions considered at Nature Cell Biology.

Version 0:

Decision Letter:

Our ref: NCB-A53441-T

6th April 2024

Dear Dr. Barr,

Thank you for submitting your revised manuscript "MDM2 acts as a timer reporting the length of mitosis" (NCB-A53441-T). It has now been seen by an arbitrator referee who provided comments on the revisions to the concerns raised by referee#2 on the previous round. Their comments are below. The reviewers find that the paper has improved in revision, and therefore we'll be happy in principle to publish it in Nature Cell Biology, pending minor revisions to satisfy the referees' final requests and to comply with our editorial and formatting guidelines. Please note that we require the referee's points to be addressed a revised version of the manuscript. Please highlight the textual changes made so that they can be easily seen in the revised version.

We are now performing detailed checks on your paper and will send you a checklist detailing our editorial and formatting requirements in about a week.

****Please do not upload the final materials and make any revisions (including the ones highlighted above) until you receive this additional information from us**.**

Please note that all key data should be presented in the main figures, with Extended Data only presenting supportive information. Please convert all Supplementary Figures into Extended Data Figures. There is a limit of 6-8 display items for articles and all key data should be presented in these figures. Please keep the current structure of the figures, but ensure that you are using the full A4 space (180mm wide x 200mm high) and increase panel size and font size (at least 7 pt) for improved readability.

Thank you again for your interest in Nature Cell Biology Please do not hesitate to contact me if you have any questions.

With best wishes,

Sabrya Carim, PhD
(she/her/hers)
Associate Editor, Nature Cell Biology
Nature Portfolio

Springer Nature
The Campus, 4 Crinan Street, London N1 9XW, UK
sabrya.carim@springernature.com
<https://orcid.org/0000-0001-9485-1938>

Reviewer #4 (Remarks to the Author):

In my view the authors have addressed the key experimental concerns of all the referees. Figures 4 to 6 do test the quantitative model of MDM2 using both population and single cell analyses. The MDM-224 treatment reduces MDM2 levels, and the authors increase the level of MDM2 to 5x endogenous in Fig 6. The experiment to compare MD-224 to Nutlin-3A (MDM2 level vs activity) is particularly informative.

That said, Referee2 is correct that there are several prior papers that propose a timer (work from Kip Sluder in particular) and narrow down the main components to p53, USP28 and 53BP-1 (work from the Desai/Oegema, Holland and Compton labs). Most of these papers are cited, though the authors should cite and discuss work from Paul Clarke proposing that MCL1 is also a mitotic timer.

The authors should also rewrite the text to make clear that their work builds on previous papers. As it currently reads, the text gives the impression that the authors carried out this work ab initio. This does not undermine the importance of the current study that clarifies a mechanism safeguarding genomic stability.

Version 1:

Decision Letter:

27th June 2024

*Please delete the link to your author homepage if you wish to forward this email to co-authors.

Dear Professor Barr,

Thank you for your patience while we have been assessing the reproducibility in your manuscript "MDM2 functions as a timer reporting the length of mitosis". Thank you also for providing clarification on which 'n' number was used to perform statistical analyses throughout the manuscript and for indicating the number of independent biological experiments performed.

I have now discussed the matter in depth with the Chief Editor. In light of these clarifications, we note that the following panels indicate that data have been derived from only 1 biological experiment (n=1 experiment): Figures 1c, 2e, 3e, 3f,g, 5b-e, ED1b, ED2c, ED6a-b, ED7b, ED7e, ED8c, ED9c,d, ED10b.

After assessing the data-set in detail, we have concluded that as a large number of figure panels display data from experiments having only been performed (in most cases) once or twice independently, the robustness of the dataset does not meet the standard we would expect in our publications. We also note that during the peer-review process referees and editors were not provided with the information about the statistical analyses to adequately evaluate the significance of the results.

We have now discussed this matter with reviewer #4 and provided them with the revised manuscript and data. The reviewer expressed serious concerns around interpretation of data from key experiments performed only once, even when there is a clear difference between control and experimental. In their view, they would not have confidence in experiments performed only one time, and they would recommend three repeats at a minimum.

We have therefore concluded that we are unable to proceed with the acceptance and publication of the current version of the manuscript.

We would be open to reconsidering the manuscript after convincing demonstration of reproducibility. We advise that the experiments be repeated at least three times. Any decision to proceed with such a revised manuscript would depend on further assessment of these data editorially and would involve reinitiating the peer-review process to determine whether the level of support for the interpretation of the data and the conclusions drawn remains unaltered.

We appreciate that this decision might come as a surprise and that this is not a trivial request. As you may know, we check all our manuscripts for compliance with our policies prior to acceptance, and we want to be consistent editorially across manuscripts. Through our editorial policies (<https://www.nature.com/nature-portfolio/editorial-policies/reporting-standards>) and referee consultations, we strive to ensure completeness of reporting and experimental reproducibility as per the field's standards. This is in our view essential to our mission as a journal.

- ensure that it conforms to our format instructions and publication policies (see below and <https://www.nature.com/nature/for-authors>).

- provide the completed Reporting Summary (found here <https://www.nature.com/documents/nr-reporting-summary.pdf>). This is essential for reconsideration of the manuscript will be available to editors and referees in the event of peer review. For more information see <http://www.nature.com/authors/policies/availability.html> or contact me.

Nature Cell Biology is committed to improving transparency in authorship. As part of our efforts in this direction, we are now requesting that all authors identified as 'corresponding author' on published papers create and link their Open Researcher and Contributor Identifier (ORCID) with their account on the Manuscript Tracking System (MTS), prior to acceptance. ORCID helps the scientific community achieve unambiguous attribution of all scholarly contributions. You can create and link your ORCID from the home page of the MTS by clicking on 'Modify my Springer Nature account'. For more information please visit www.springernature.com/orcid.

This journal strongly supports public availability of data. Please place the data used in your paper into a public data repository, or alternatively, present the data as Supplementary Information. If data can only be shared on request, please explain why in your Data Availability Statement, and also in the correspondence with your editor. Please note that for some data types, deposition in a public repository is mandatory - more information on our data deposition policies and available repositories appears below.

Please submit the revised manuscript files using this link:

Link Redacted

We hope that you will find our editorial guidance helpful. Please do not hesitate to contact us if there is anything you would like to discuss.

Best wishes,

Sabrya Carim

Sabrya Carim, PhD
(she/her/hers)
Associate Editor, Nature Cell Biology
Nature Portfolio

Springer Nature
The Campus, 4 Crinan Street, London N1 9XW, UK
sabrya.carim@springernature.com
<https://orcid.org/0000-0001-9485-1938>

READABILITY OF MANUSCRIPTS – Nature Cell Biology is read by cell biologists from diverse backgrounds, many of whom are not native English speakers. Authors should aim to communicate their findings clearly, explaining technical jargon that might be unfamiliar to non-specialists, and avoiding non-standard abbreviations. Titles and abstracts should concisely

communicate the main findings of the study, and the background, rationale, results and conclusions should be clearly explained in the manuscript in a manner accessible to a broad cell biology audience. Nature Cell Biology uses British spelling.

Methods should be written concisely, but should contain all elements necessary to allow interpretation and replication of the results. As a guideline, Methods sections typically do not exceed 3,000 words. The Methods should be divided into subsections listing reagents and techniques. When citing previous methods, accurate references should be provided and any alterations should be noted. Information must be provided about: antibody dilutions, company names, catalogue numbers and clone numbers for monoclonal antibodies; sequences of RNAi and cDNA probes/primers or company names and catalogue numbers if reagents are commercial; cell line names, sources and information on cell line identity and authentication. Animal studies and experiments involving human subjects must be reported in detail, identifying the committees approving the protocols. For studies involving human subjects/samples, a statement must be included confirming that informed consent was obtained. Statistical analyses and information on the reproducibility of experimental results should be provided in a section titled "Statistics and Reproducibility".

All Nature Cell Biology manuscripts submitted on or after March 21 2016 must include a Data availability statement as a separate section after Methods but before references, under the heading "Data Availability". For Springer Nature policies on data availability see <http://www.nature.com/authors/policies/availability.html>; for more information on this particular policy see <http://www.nature.com/authors/policies/data/data-availability-statements-data-citations.pdf>. The Data availability statement should include:

- Accession codes for primary datasets (generated during the study under consideration and designated as "primary accessions") and secondary datasets (published datasets reanalysed during the study under consideration, designated as "referenced accessions"). For primary accessions data should be made public to coincide with publication of the manuscript. A list of data types for which submission to community-endorsed public repositories is mandated (including sequence,

structure, microarray, deep sequencing data) can be found here <http://www.nature.com/authors/policies/availability.html#data>.

- Unique identifiers (accession codes, DOIs or other unique persistent identifier) and hyperlinks for datasets deposited in an approved repository, but for which data deposition is not mandated (see here for details <http://www.nature.com/sdata/data-policies/repositories>).
- At a minimum, please include a statement confirming that all relevant data are available from the authors, and/or are included with the manuscript (e.g. as source data or supplementary information), listing which data are included (e.g. by figure panels and data types) and mentioning any restrictions on availability.
- If a dataset has a Digital Object Identifier (DOI) as its unique identifier, we strongly encourage including this in the Reference list and citing the dataset in the Methods.

We recommend that you upload the step-by-step protocols used in this manuscript to protocols.io. More details can be found at <https://www.protocols.io/help/publish-articles>.

All imaging data should be accompanied by scale bars, which should be defined in the legend.

Cropped images of gels/blots are acceptable, but need to be accompanied by size markers, and to retain visible background signal within the linear range (i.e. should not be saturated). The boundaries of panels with low background have to be demarcated with black lines. Splicing of panels should only be considered if unavoidable, and must be clearly marked on the figure, and noted in the legend with a statement on whether the samples were obtained and processed simultaneously. Quantitative comparisons between samples on different gels/blots are discouraged; if this is unavoidable, it should only be performed for samples derived from the same experiment with gels/blots were processed in parallel, which needs to be stated in the legend.

The total number of Supplementary Figures (not including the “unprocessed scans” Supplementary Figure) should not exceed the number of main display items (figures and/or tables (see our Guide to Authors and March 2012 editorial <http://www.nature.com/nature/authors/submit/index.html#supinfo>; <http://www.nature.com/nature/journal/v14/n3/index.html#ed>). No restrictions apply to Supplementary Tables or Videos, but we advise authors to be selective in including supplemental data.

GUIDELINES FOR EXPERIMENTAL AND STATISTICAL REPORTING

REPORTING REQUIREMENTS – We are trying to improve the quality of methods and statistics reporting in our papers. To that end, we are now asking authors to complete a reporting summary that collects information on experimental design and reagents. The Reporting Summary can be found here <https://www.nature.com/documents/nr-reporting-summary.pdf>. If you would like to reference the guidance text as you complete the template, please access these flattened versions at <http://www.nature.com/authors/policies/availability.html>.

We strongly recommend the presentation of source data for graphical and statistical analyses as a separate Supplementary Table, and request that source data for all independent repeats are provided when representative experiments of multiple independent repeats, or averages of two independent experiments are presented. This supplementary table should be in Excel format, with data for different figures provided as different sheets within a single Excel file. It should be labelled and numbered as one of the supplementary tables, titled “Statistics Source Data”, and mentioned in all relevant figure legends.

Version 2:

Decision Letter:

Our ref: NCB-A53441B

25th September 2024

Dear Professor Barr,

Thank you for submitting your revised manuscript "MDM2 functions as a timer reporting the length of mitosis" (NCB-A53441B). It has now been seen by the arbitrator referee and their comments are below. The reviewer finds that the paper has improved in revision, and therefore we'll be happy in principle to publish it in Nature Cell Biology, pending minor revisions to comply with our editorial and formatting guidelines.

We ask that you ensure that all figures are arranged in portrait format to adhere to a maximum page size of roughly 180mm wide x 200mm high and use a font size of no smaller than 7pt throughout for all main and extended data figures.

We are now performing detailed checks on your paper and will send you a checklist detailing our editorial and formatting requirements in about 1-2 weeks. Please do not upload the final materials and make any revisions until you receive this additional information from us.

Thank you again for your interest in Nature Cell Biology Please do not hesitate to contact me if you have any questions.

Best,

Sabrya Carim, PhD
(she/her/hers)
Associate Editor, Nature Cell Biology
Nature Portfolio

Springer Nature
The Campus, 4 Crinan Street, London N1 9XW, UK
sabrya.carim@springernature.com
<https://orcid.org/0000-0001-9485-1938>

Reviewer #4 (Remarks to the Author):

The authors have performed an appropriate number of independent experiments and the results support their conclusions.

Version 3:

Decision Letter:

Dear Dr Barr,

I am pleased to inform you that your manuscript, "MDM2 functions as a timer reporting the length of mitosis", has now been accepted for publication in Nature Cell Biology.

Please note that *Nature Cell Biology* is a Transformative Journal (TJ). Authors may publish their research with us through the traditional subscription access route or make their paper immediately open access through payment of an article-processing charge (APC). Authors will not be required to make a final decision about access to their article until it has been accepted. [Find out more about Transformative Journals](https://www.springernature.com/gp/open-research/transformative-journals)

If you have not already done so, we strongly recommend that you upload the step-by-step protocols used in this manuscript to protocols.io (<https://protocols.io>), an open online resource that allows researchers to share their detailed experimental know-how. All uploaded protocols are made freely available and are assigned DOIs for ease of citation. Protocols and Nature Portfolio journal papers in which they are used can be linked to one another, and this link is clearly and prominently visible in the online versions of both. Authors who performed the specific experiments can act as primary authors for the Protocol as they will be best placed to share the methodology details, but the Corresponding Author of the present research paper should be included as one of the authors. By uploading your Protocols onto protocols.io, you are enabling researchers to more readily reproduce or adapt the methodology you use, as well as increasing the visibility of your protocols and papers. You can also establish a dedicated workspace to collect your lab Protocols. Further information can be found at <https://www.protocols.io/help/publish-articles>.

Nature Cell Biology encourages authors presenting evidence for cell, biological, molecular, and genetic interactions to

consider communicating these findings using Biofactoid (<https://biofactoid.org/>). This tool helps users share a searchable representation of interactions (e.g. binding, gene expression, post-translational modification) between genes, gene products, or chemicals. Information added to Biofactoid, with author attribution, is shared on social media and public databases, such as Pathway Commons, where it can be discovered and analyzed in the context of a large and growing corpus of knowledge.

With kind regards,

Sabrya Carim, PhD
(she/her/hers)
Senior Editor, Nature Cell Biology
Nature Portfolio

Springer Nature
The Campus, 4 Crinan Street, London N1 9XW, UK
sabrya.carim@springernature.com
<https://orcid.org/0000-0001-9485-1938>

** Visit the Springer Nature Editorial and Publishing website at http://editorial-jobs.springernature.com?utm_source=ejp_NCB_email&utm_medium=ejp_NCB_email&utm_campaign=ejp_NCB for more information about our career opportunities. If you have any questions please click [here](mailto:editorial.publishing.jobs@springernature.com).

Our ref: NCB-A53441-T

6th April 2024

Dear Dr. Barr,

Thank you for submitting your revised manuscript "MDM2 acts as a timer reporting the length of mitosis" (NCB-A53441-T). It has now been seen by an arbitrator referee who provided comments on the revisions to the concerns raised by referee#2 on the previous round. Their comments are below. The reviewers find that the paper has improved in revision, and therefore we'll be happy in principle to publish it in Nature Cell Biology, pending minor revisions to satisfy the referees' final requests and to comply with our editorial and formatting guidelines. Please note that we require the referee's points to be addressed a revised version of the manuscript. Please highlight the textual changes made so that they can be easily seen in the revised version.

We are now performing detailed checks on your paper and will send you a checklist detailing our editorial and formatting requirements in about a week.

****Please do not upload the final materials and make any revisions (including the ones highlighted above) until you receive this additional information from us**.**

Please note that all key data should be presented in the main figures, with Extended Data only presenting supportive information. Please convert all Supplementary Figures into Extended Data Figures. There is a limit of 6-8 display items for articles and all key data should be presented in these figures. Please keep the current structure of the figures, but ensure that you are using the full A4 space (180mm wide x 200mm high) and increase panel size and font size (at least 7 pt) for improved readability.

Thank you again for your interest in Nature Cell Biology Please do not hesitate to contact me if you have any questions.

With best wishes,

Sabrya Carim, PhD
(she/her/hers)
Associate Editor, Nature Cell Biology
Nature Portfolio

Springer Nature
The Campus, 4 Crinan Street, London N1 9XW, UK
sabrya.carim@springernature.com
<https://orcid.org/0000-0001-9485-1938>

Reviewer #4 (Remarks to the Author):

In my view the authors have addressed the key experimental concerns of all the referees. Figures 4 to 6 do test the quantitative model of MDM2 using both population and single cell analyses. The MDM-224 treatment reduces MDM2 levels, and the authors increase the level of MDM2 to 5x endogenous in Fig 6. The experiment to compare MD-224 to Nutlin-3A (MDM2 level vs activity) is particularly informative.

Response: we thank the reviewer for their detailed reading and critical appraisal of our manuscript.

That said, Referee2 is correct that there are several prior papers that propose a timer (work from Kip Sluder in particular) and narrow down the main components to p53, USP28 and 53BP-1 (work from the Desai/Oegema, Holland and Compton labs). Most of these papers are cited, though the authors should cite and discuss work from Paul Clarke proposing that MCL1 is also a mitotic timer.

Response: The work from the Clarke lab relates to apoptosis rather than G1 cell cycle arrest which is the focus of our work, so we did not include this in our original text. We have added text to mention this and have now cited work from the Clarke lab in the discussion (see lines 299-301).

We have revised the discussion text as follows:

These findings support the conclusion that MDM2, due to its short half-life, is a key timer component in the mechanism that triggers a robust cell cycle arrest in the G1 following a prolonged delay in mitosis ^{21, 22}. This is reminiscent of the role played by the anti-apoptotic protein Mcl-1 in the regulation of apoptosis after extended mitosis. In that case, slow APC/C-dependent destruction of Mcl-1 acts as a timer for apoptosis ^{32, 33}. By contrast, MDM2's timer properties arise through a self-catalysed ubiquitination and proteasomal destruction mechanism, and the attenuation of protein synthesis in mitosis.

The authors should also rewrite the text to make clear that their work builds on previous papers. As it currently reads, the text gives the impression that the authors carried out this work ab initio. This does not undermine the importance of the current study that clarifies a mechanism safeguarding genomic stability.

Response: This view was not shared by the other 3 reviewers who previously evaluated the work. To fully review the topic is beyond the scope of a brief introductory paragraph. However, we do our best to describe and cite the relevant previous work in both the introductory paragraph and discussion sections. It was perhaps unclear that previous work had not examined normal conditions, which we now clarify (see introduction and highlighted text below).

We have revised the introduction text as follows:

Chromosome instability, aneuploidy, the removal of centrosomes, or anti-mitotic drugs targeting the microtubule cytoskeleton delay progression through mitosis, triggering a p53 and p21 dependent cell cycle arrest in G1 thought to prevent proliferation of damaged cells ¹⁻⁶. This response is lost in cancer cells in which p53 has become inactivated by mutation or other mechanisms, including expression of

viral oncoproteins⁷⁻¹⁰. Crucially, G1 cell cycle arrest following prolonged mitosis occurs even in the absence of detectable DNA damage, suggesting it has a different cause, proposed to be a direct consequence of the increased time spent in mitosis¹⁻¹⁰. Because most studies have focussed on drug-induced delays to mitosis, the role of this pathway for normal cell function remains unclear. Therefore, we asked if variation in the length of mitosis, inherent in the stochastic search-capture process underpinning chromosome alignment even in the absence of any perturbation, influences the behaviour of untransformed diploid cells with wild-type p53 (p53^{WT}) in the ensuing G1.

NCB-A53441-T – MDM2 functions as a timer reporting the length of mitosis

Detailed list of revisions:

Figure 1a-1c. Experiments analysing the stochastic variation in the length of mitosis and frequency of G1 arrest and role for p53. This is compiled from the analysis of multiple cells imaged in 3 independent experiments. Representative images are shown in Fig 1a. Mean time in mitosis is shown in violin plots in Fig. 1b for p53^{WT} and p53^{KO} cell lines. The proportion of cells arresting in G1 for each category is indicated in the figure below the plot in Figure 1c (mean \pm s.d.; n = 3 independent experiments). Full details of cell numbers analysed and statistical methods used are provided in the figure legend.

Figure 1d. Cell cycle arrest behaviour after mitotic delay in p53^{WT} and p53^{KO} hTERT-RPE1 cells. This data is from 8 independent experiments with similar results. Representative images of cell growth are shown from one of these experiments, and the line graph shows relative cell growth (mean \pm s.e.m; n = 8 independent experiments).

Figure 2e (previously Fig.2g). Proteasome inhibition blocks mitotic MDM2 destruction. A bar graph showing MDM2 levels has been added to the figure (mean \pm s.d.; n = 3 independent experiments), alongside a representative Western blot from one of the 3 independent experiments.

Figure 2g (previously Fig. 2e). Pulse-labelling with ³⁵S-methionine to investigate global protein synthesis and MDM2 synthesis in mitosis compared to interphase cell population. The data shown in the gels is a representative example taken from 3 independent experiments. Two bar graphs have been added to the figure panel showing the level of synthesis as mean \pm s.d., n = 3 independent experiments, for total protein and MDM2, respectively. Full details of statistical methods used are provided in the figure legend.

Figure 3c, 3d and 3e. Analysis of MDM2 stability. The data shown in each panel are representative of 3 independent experiments giving the same outcome. For Fig. 3d the level of MDM2 after proteasome inhibition was measured and plotted in a line graph as a function of time as mean \pm s.d., n = 3 independent experiments. For Fig. 3e the level of MDM2 in interphase and mitotic cells following UBE2D depletion was measured and plotted in bar graphs as mean \pm s.d., n = 3 independent experiments. Full details of statistical methods used are provided in the figure legend.

Figure 3f and 3g. Effect of UBE2D depletion on cell cycle progression. This is compiled from the analysis of multiple cells imaged in 3 independent experiments. Representative images are shown in Fig. 3f. The proportion of cells arresting in G1 for each category is indicated in bar graphs in Fig. 3g, mean \pm s.d., n = 3 independent experiments. Full details of cell numbers analysed and statistical methods used are provided in the figure legend.

Figure 5a-5d. Effect of mitotic delays and manipulation of MDM2 stability on cell cycle progression in p53^{WT} and p53^{KO} cell lines. This is compiled from the analysis of multiple cells imaged for the 10 different conditions in 3 independent experiments (30 independent experiments in total). Representative images are shown in Extended Data Fig 8a-8c for these different conditions. Mean time spent in G1 for cells entering S-phase is shown in box and whisker plots above each cell cycle history plot. The new panel Fig. 5d shows the proportion

NCB-A53441-T – MDM2 functions as a timer reporting the length of mitosis

of cells arresting in G1 for each category shown in Fig. 5a-5c as mean \pm s.d., $n = 3$ independent experiments. Full details of cell numbers analysed and statistical methods used are provided in the figure legend.

Figure 5e (previously Fig. 5d). Effect of MDM2 destruction in G1/S/G2 phases. This is compiled from the analysis of multiple cells imaged in 3 independent experiments. No cells in any of the 3 independent experiments showed G1 arrest behaviour, indicated in the numbers shown at the bottom of the panel.

Figure 5f (previously Fig. 5e). Effect of mitotic delays and MDM2 destruction in mitosis on p21 levels in the following G1 phase. Induction of p21-GFP in G1 was tracked in multiple cells in 3 independent experiments, and is plotted in the line graph for each condition as mean \pm s.e.m., $n = 3$ independent experiments. Full details of cell numbers analysed are provided in the figure legend.

Figure 5g (previously Fig. 5f). Measurement of the MDM2 threshold for the mitotic timer effect and G1 arrest. This data is from 3 independent experiments. The Western blot is a representative example, and the line graphs show MDM2 level measured by Western blotting, and nuclear p21 intensity and the proportion of cells in G1 analysed from immunofluorescence as a function of MD-224 concentration, mean \pm s.e.m., $n = 3$ independent experiments. Full details are provided in the figure legend.

Figure 6b. Effect of MDM2 overexpression on the mitotic timer threshold. This data is from 3 independent experiments. The line graph shows nuclear p21 intensity, mean \pm s.e.m., $n = 3$ independent experiments, for parental and MDM2 overexpressing cells. The inset bar graph shows a quantification of MDM2 protein levels in both cell lines, mean \pm s.d., $n = 3$ independent experiments.

Ext Data Figure 1. Extended characterisation of hTERT-RPE1 FUCCI p53^{WT} and p53^{KO} cell lines in interphase and mitosis. Representative image data for 3 independent experiments is shown in Ext Data Fig 1a and 1b for centriole/centrosome number and cilium formation, respectively. The bar graph in Ext Data Fig 1b shows cilium formation, mean \pm s.e.m., $n = 3$ independent experiments. Full details of cell numbers analysed and statistical methods used are provided in the figure legend. For the analysis of mitosis in Ext Data Fig. 1c and 1d, representative images taken from one of 3 independent experiments are shown. The stacked bar graph in Ext Data Fig 1d shows the analysis of cell cycle progression from mitosis into anaphase and G1, mean \pm s.d., $n = 3$ independent experiments.

Ext Data Figure 2. Analysis of DNA damage in interphase and mitosis in hTERT-RPE1 FUCCI p53^{WT} cells. This figure was revised to provide a more complete analysis of DNA damage in hTERT-RPE1 cells exiting delayed mitosis, and displays the behaviour of 3 independent experiments. A representative Western blot analysis of control/resting state and drug-induced DNA damage in asynchronous culture and in cells released from a mitotic arrest is shown in Ext Data Fig. 2a with corresponding DNA damage marker measurements in bar graphs, mean \pm s.d., $n = 3$ independent experiments, in Ext Data Fig. 2b and 2c. Representative image data showing the cell cycle response to DNA damage compared to control cells for 3 independent experiments is shown in Ext Data Fig. 2d, with a scatter plot

NCB-A53441-T – MDM2 functions as a timer reporting the length of mitosis

showing replicates and mean \pm s.d. for the three independent experimental replicates in Ext Data Fig. 2e (G1 cells).

Ext Data Figure 3. Analysis of MDM2 in different cancer cell lines. Western blots in Ext Data Fig. 3a and 3c are representative examples from 3 independent experiments, with corresponding bar graphs showing MDM2 levels, mean \pm s.e.m., $n = 3$ independent experiments, in Ext Data Fig. 3b and 3d, respectively. The representative Western blot in Ext Data Fig 3a has been replaced with a more evenly loaded replicate.

Ext Data Figure 6. Role of APC/C and SCF in the regulation of MDM2 stability. Western blots in Ext Data Fig. 6a-6c are representative examples from 3 independent experiments, with corresponding bar graphs or line graphs showing MDM2 levels, mean \pm s.e.m., $n = 3$ independent experiments. The bar graphs in Ext Data Fig. 6d show the RT-qPCR measurement of β -TCRP1 and β -TRCP-2 mRNA levels from 2 independent biological replicates, each with 3 technical replicates to confirm successful knockdown of β -TrCP2, for which we were unable to find an effective antibody to use in Western blotting. Representative images of the effect of transient proteasome inhibition at low and high doses on mitotic exit are shown in Ext Data Fig. 6e, with a bar graph in Ext Data Fig. 6f, mean \pm s.d., $n = 3$ independent experiments.

Ext Data Figure 7. Controls for the MD-224 MDM2-directed PROTAC. All Western blots shown in the figure are representative examples from 3 independent experiments, with bar graphs for the levels of MDM2 under the different conditions as mean \pm s.d., $n = 3$ in Ext Data Fig. 7b,d and f. A line graph showing recovery of MDM2 protein levels following MD-224 treatment in p53^{WT} and p53^{KO} cells is plotted in Ext Data Fig 7c (mean \pm s.e.m., $n = 3$ independent experiments). The representative Western blot in Ext Data Fig 7c has been replaced with a more evenly loaded replicate.

Ext Data Figure 9. Activation of the mitotic timer using CENPE inhibition. The effects of mitotic delays caused by different agents are compared in Western blots representative of 3 independent experiments in Ext Data Fig. 9a with corresponding bar graphs for two different MDM2 antibodies in Ext Data Fig. 9b, mean \pm s.d., $n = 3$ independent experiments. The effects of mitotic delays caused by CENPE inhibition on cell cycle progression are shown in the representative Western blot and image data in Ext Data Fig. 9c and 9d taken from 3 independent experiments. These same 3 experiments were quantified and the data presented in the cell cycle history plots and bar graphs of G1-S transition, mean \pm s.d., $n = 3$ independent experiments, in Ext Data Fig. 9e. G1 length in cells with G1/S transition is plotted in a box and whiskers blot in Ext Data Fig. 9f taken from 3 independent experiments. Full details of statistical methods used are provided in the figure legend.

Extended Data Figure 10. Effect of MDM2 overexpression on MDM2 half-life and the p21 threshold for p21 induction. All Western blots shown in the figure are representative examples from 3 independent experiments, with line graphs for the levels of MDM2 under the different conditions plotted mean \pm s.e.m., $n = 3$ independent experiments.